# Generalised Bayesian Filtering via Sequential Monte Carlo

**Ayman Boustati**[*]
University of Warwick
a.boustati@warwick.ac.uk

**Ömer Deniz Akyildiz**[*]
The Alan Turing Institute
University of Warwick
omer.akyildiz@warwick.ac.uk

**Theodoros Damoulas**
The Alan Turing Institute
University of Warwick
t.damoulas@warwick.ac.uk

**Adam M. Johansen**
University of Warwick
The Alan Turing Institute
a.m.johansen@warwick.ac.uk

## Abstract

We introduce a framework for inference in general state-space hidden Markov models (HMMs) under likelihood misspecification. In particular, we leverage the loss-theoretic perspective of Generalized Bayesian Inference (GBI) to define generalised filtering recursions in HMMs, that can tackle the problem of inference under model misspecification. In doing so, we arrive at principled procedures for robust inference against observation contamination by utilising the $\beta$-divergence. Operationalising the proposed framework is made possible via sequential Monte Carlo methods (SMC), where most standard particle methods, and their associated convergence results, are readily adapted to the new setting. We apply our approach to object tracking and Gaussian process regression problems, and observe improved performance over both standard filtering algorithms and other robust filters.

## 1  Introduction

Estimating the hidden states in dynamical systems is a long-standing problem in many fields of science and engineering. This can be formulated as an inference problem of a general state-space hidden Markov model (HMM) defined via two processes, *the hidden process* $(\mathbf{x}_t)_{t \geq 0}$, and *the observation process* $(\mathbf{y}_t)_{t \geq 1}$. More precisely, we consider the general state-space hidden Markov models of the form

$$\mathbf{x}_0 \sim \pi_0(\mathbf{x}_0), \quad (1) \qquad \mathbf{x}_t | \mathbf{x}_{t-1} \sim f_t(\mathbf{x}_t | \mathbf{x}_{t-1}), \quad (2) \qquad \mathbf{y}_t | \mathbf{x}_t \sim g_t(\mathbf{y}_t | \mathbf{x}_t), \quad (3)$$

where $\mathbf{x}_t \in \mathsf{X}$ for $t \geq 0$, $y_t \in \mathsf{Y}$ for $t \geq 1$, $f_t$ is a Markov kernel on $\mathsf{X}$ and $g_t : \mathsf{Y} \times \mathsf{X} \to \mathbb{R}_+$ is the likelihood function. We assume $\mathsf{X} \subseteq \mathbb{R}^{d_x}$ and $\mathsf{Y} \subseteq \mathbb{R}^{d_y}$ for convenience; however, the extension to general Polish spaces follows directly. The key inference problem in this model class is estimating is the *filtering distributions*, i.e. the posterior distributions of the hidden states $(\mathbf{x}_t)_{t \geq 0}$ given the observations $\mathbf{y}_{1:t}$ denoted as $(\pi_t(\mathbf{x}_t | \mathbf{y}_{1:t}))_{t \geq 1}$ — commonly known as *Bayesian filtering* [1, 2].

Under assumptions of linearity and Gaussianity, the inference problem for the hidden states of HMMs can be solved analytically via the Kalman filter [3]. However, inference for general HMMs of the form (1)–(3) with nonlinear, non-Gaussian transitions and likelihoods lacked a general, principled solution until the arrival of the particle filtering schemes [4]. Particle filters (PFs) have become ubiquitous for

---

[*]Equal contribution.

Bayesian filtering in the general setting. In short, the PFs retain a weighted collection of Monte Carlo samples representing the filtering distribution $\pi_t(\mathbf{x}_t|\mathbf{y}_{1:t})$ and recursively approximate the sequence of distributions $(\pi_t)_{t \geq 0}$ using a particle mutation-selection scheme [5].

While PFs (and other inference schemes for HMMs) implicitly assume that the assumed model is well-specified, it is important to consider whether the proposed model class includes the true data-generating mechanism (DGM). In particular, for general state-space HMMs, misspecification can occur if the true dynamics of the hidden process significantly differ from the assumed model $f_t$, or if the true observation model is markedly different from the assumed likelihood model $g_t$, e.g. corruption by heavy tailed noise. The latter case is of widespread interest within the field of *robust statistics* [6] and has recently attracted significant interest in the machine learning community [7]. It is the setting that this paper seeks to address.

When the true DGM cannot be modelled, one principled approach to address misspecification is Generalized Bayesian Inference (GBI) [8]. This approach views classical Bayesian inference as a loss minimisation procedure in the space of probability measures, a view first developed by [9]. In particular, the standard Bayesian update can be derived from this view, where a loss function is constructed using the Kullback-Leibler (KL) divergence from the empirical distribution of the observations to the assumed likelihood [8]. The KL divergence is sensitive to outliers [10], hence the overall inference procedure is not robust to observations that are incompatible with the assumed model. A principled remedy is to replace the KL divergence with alternative discrepancy, such as the $\beta$-divergence, which makes the overall procedure more robust [11] while retaining interpretability.

Previous work on robust particle filters have been done for handling outliers, sensor failures and misspecification of the transition model [12, 13, 14, 15, 16, 17, 18, 19]. However, these approaches are either based on problem-specific heuristic outlier detection schemes, or make strong assumptions about the DGM in order to justify the use of heavy-tailed distributions [15]. This requires knowledge of the contamination mechanism that is implicitly embedded in the likelihood. Thus, this work considers the challenging M-open settings: we do not assume access to a family of models which includes the true generative model. This is qualitatively different from classical parameter estimation approaches [20]; consequently, model selection schemes cannot generally be used to correct for misspecification (note the additional complications associated with parameter estimation in misspecified scenarios, see, e.g. [21]: not only does estimating parameters not address misspecification, but even the interpretation of estimated parameters is difficult). Furthermore, this case is not addressed by sequential Monte Carlo (SMC) algorithms under model uncertainty (see, e.g., [22]) where the true model is assumed to be available among many candidate models. For instance in [22], information from many candidate models is fused according to their predictive performance, which is a pragmatic solution with good empirical performance when a good suite of candidates is available. In contrast, we assume that we do not have any access at all to the true underlying generating mechanism.

In this work we propose a principled approach to robust filtering that does not impose additional modelling assumptions. We adapt the GBI approach of [8] to the Bayesian filtering setting and develop sequential Monte Carlo methods for inference. We illustrate the performance of this approach, using the $\beta$-divergence, to mitigate the effect of outliers. We show that this approach significantly improves the PF performance in settings with contaminated data, while retaining a general and principled approach to inference. We provide empirical results that demonstrate improvement over Kalman and particle filters for both linear and non-linear HMMs. We further provide comparisons with various robust schemes against heavy-tailed noise, including t-based likelihoods [15] or auxiliary particle filters (APFs) [12]. Finally, exploiting the state-space representations of Gaussian processes (GPs) [23], we demonstrate our framework on London air pollution data using robust GP regression which has linear time-complexity in the number of observations.

**Notation.** We denote the space of bounded, Borel measurable functions on $\mathsf{X}$ as $B(\mathsf{X})$. We denote the Dirac measure located at $y$ as $\delta_{\mathbf{y}}(\mathrm{d}\mathbf{x})$ and note that $f(\mathbf{y}) = \int f(\mathbf{x})\delta_{\mathbf{y}}(\mathrm{d}\mathbf{x})$ for $f \in B(\mathsf{X})$. We denote the Borel subsets of $\mathsf{X}$ as $\mathcal{B}(\mathsf{X})$ and the set of probability measures on $(\mathsf{X}, \mathcal{B}(\mathsf{X}))$ as $\mathcal{P}(\mathsf{X})$. For a probability measure $\mu \in \mathcal{P}(\mathsf{X})$ and $\varphi \in B(\mathsf{X})$, we write $\mu(\varphi) := \int \varphi(\mathbf{x})\mu(\mathrm{d}\mathbf{x})$. Given a probability measure $\mu$, we abuse the notation denoting its density with respect to the Lebesgue measure as $\mu(\mathbf{x})$.

## 2 Background

### 2.1 Generalized Bayesian Inference (GBI)

Bayesian inference implicitly assumes that the generative model is well-specified, in particular, the observations are generated from the assumed likelihood model. When this assumption is not expected to hold in real-world scenarios, one may wish to take into account the discrepancy between the true DGM and the assumed likelihood. GBI [8] is an approach to deal with such cases. Here, we present the main idea of GBI and refer the reader to the appendix for a more detailed description and to the original reference for a full-treatment.

For the simple Bayesian updating setup, consider a prior $\pi_0$ and the assumed likelihood function $g(\mathbf{y}|\mathbf{x})$. The posterior $\pi(\mathbf{x}|\mathbf{y}) =: \pi(\mathbf{x})$ is given by Bayes rule $\pi(\mathbf{x}) = \pi_0(\mathbf{x})\frac{g(\mathbf{y}|\mathbf{x})}{Z}$, where $Z := \int g(\mathbf{y}|\mathbf{x})\pi_0(\mathbf{x})\mathrm{d}\mathbf{x}$. [9] and [8] showed that this update can be seen as a special case of a more general update rule, which can be described as a solution of an optimisation problem in the space of measures. This leads to a more general belief updating rule given by

$$\pi(\mathbf{x}) = \pi_0(\mathbf{x})\frac{G(\mathbf{y}|\mathbf{x})}{Z}, \tag{4}$$

with $G(\mathbf{y}|\mathbf{x}) := \exp(-\ell(\mathbf{x},\mathbf{y}))$ where $\ell(\mathbf{x},\mathbf{y})$ is a loss function connecting the observations to the model parameters. Specifying $\ell(\mathbf{x},\mathbf{y})$ as the cross-entropy (from the KL-divergence) of the assumed likelihood relative to the empirical distribution of the data recovers the standard Bayes update.

As noted before, the standard Bayes update is not robust to outliers due to the properties of KL divergence [10]. Hence, substituting the cross-entropy with a more robust loss such as the $\beta$-cross-entropy [7], based on the $\beta$-divergence, can make the inference more robust. Specifically, in this setting the generalised Bayes update for the likelihood $g(\mathbf{y}|\mathbf{x})$ is written as $\pi(\mathbf{x}) = \pi_0(\mathbf{x})\frac{G^\beta(\mathbf{y}|\mathbf{x})}{Z_\beta}$, where

$$G^\beta(\mathbf{y}|\mathbf{x}) = \exp\left(\frac{1}{\beta}g(\mathbf{y}|\mathbf{x})^\beta - \frac{1}{\beta+1}\int g(\mathbf{y}'|\mathbf{x})^{\beta+1}\mathrm{d}\mathbf{y}'\right). \tag{5}$$

One can consider $G^\beta(\mathbf{y}|\mathbf{x})$ as a generalised likelihood, resulting from the use of a different loss function compared to the standard Bayes procedure. Here $\beta$ is a hyperparameter that needs to be selected depending on the degree of misspecification. More precisely, it is a parameter of a specified loss function: a subjective (generalised) Bayesian choice characterising confidence in model correctness. In general $\beta \in (0,1)$ and $\lim_{\beta\to 0} G^\beta(\mathbf{y}|\mathbf{x}) = g(\mathbf{y}|\mathbf{x})$. Thus, intuitively, small $\beta$ values are suitable for mild model misspecification and large $\beta$ values are suitable when the assumed model is expected to significantly deviate from the true model. In the experimental section, we devote some attention to the selection of $\beta$ and sensitivity analysis.

Generalised Bayesian updating is more robust against outliers if a suitable divergence is chosen [24, 25, 10]. We note that GBI is conceptually different from approximate Bayesian methods with alternative divergences such as [26, 27, 28, 29]. While these methods target approximate posteriors that minimise some discrepancy from the true posterior and are not necessarily robust, GBI methods change the inference target from the standard Bayesian posterior (obtained by setting $\ell(\mathbf{x},\mathbf{y})$ to the KL divergence) to a different target distribution with more desirable properties such as robustness to outliers. We also remark that the qualitative behaviour of this robustness is different than simply inflating the variance of the likelihood (see Appendix B for more discussion from the perspective of influence functions). Later, we demonstrate how the GBI approach can be used to construct robust PF procedures.

### 2.2 Sequential Monte Carlo for HMMs

Let $\mathbf{x}_{1:T}$ be a hidden process with $\mathbf{x}_t \in \mathsf{X}$ and $\mathbf{y}_{1:T}$ an observation process with $\mathbf{y}_t \in \mathsf{Y}$. Our goal is to conduct inference in HMMs of the form (1)–(3) where $\pi_0(\cdot)$ is a prior probability distribution on the initial state $\mathbf{x}_0$, $f_t(\mathbf{x}|\mathbf{x}')$ is a Markov transition kernel on $\mathsf{X}$ and $g_t(\mathbf{y}_t|\mathbf{x}_t)$ is the likelihood for observation $\mathbf{y}_t$. The observation sequence $\mathbf{y}_{1:T}$ is assumed to be fixed but otherwise arbitrary.

The typical interest in probabilistic models is the estimation of expectations of general test functions with respect to the posterior distribution, in this case, of the hidden process $\pi_t(\mathbf{x}_t|\mathbf{y}_{1:t})$ and the

associated joint distributions $p_t(\mathbf{x}_{0:t}|\mathbf{y}_{1:t})$. More precisely, given a bounded test function $\varphi \in B(\mathsf{X})$, we are interested in estimating integrals of the form

$$\pi_t(\varphi) = \int \varphi(\mathbf{x}_t)\pi_t(\mathbf{x}_t|\mathbf{y}_{1:t}). \tag{6}$$

Kalman filtering [3, 1] can be used to obtain closed form expressions for $(\pi_t, p_t)_{t \geq 0}$ if $f_t$ and $g_t$ are linear-Gaussian. However, for non-linear or non-Gaussian cases, the target distributions are almost always intractable, requiring an alternative approach, such as SMC methods [5, 30]. Known as Particle Filters (PFs) when employed in the HMM setting, SMC methods combine importance sampling and resampling algorithms tailored to approximate the solution of the filtering and smoothing problems.

In a typical iteration, a PF method proceeds as follows: given a collection of samples $\{\mathbf{x}_{t-1}^{(i)}\}_{i=1}^{N}$ representing the posterior $\pi_{t-1}(\mathbf{x}_{t-1}|\mathbf{y}_{1:t-1})$, it first samples from a (possibly observation dependent) proposal $\bar{\mathbf{x}}_t^{(i)} \sim q_t(\mathbf{x}_t|\mathbf{x}_{1:t-1}^{(i)}, \mathbf{y}_{1:t})$. It then computes weights for each sample (particle) $\bar{\mathbf{x}}_{t-1}^{(i)}$ in the collection for a given observation $\mathbf{y}_t$, evaluating its fitness with respect to the likelihood $g_t$ as $\mathsf{w}_t^{(i)} \propto g_t(\mathbf{y}_t|\bar{\mathbf{x}}_t^{(i)})\frac{f_t(\bar{\mathbf{x}}_t^{(i)}|\mathbf{x}_{t-1}^{(i)})}{q_t(\bar{\mathbf{x}}_t^{(i)}|\mathbf{x}_{1:t-1}^{(i)}, \mathbf{y}_t)}$, where $\sum_{i=1}^{N} \mathsf{w}_t^{(i)} = 1$. Finally, an optional resampling step [2] is used to prevent degeneracy, leading to $\mathbf{x}_t^{(i)} \sim \sum_{i=1}^{N} \mathsf{w}_t^{(i)}\delta_{\bar{\mathbf{x}}_t^{(i)}}(\mathrm{d}\mathbf{x}_t)$. One can then construct the empirical measure $\pi_t^N(\mathrm{d}\mathbf{x}_t|\mathbf{y}_{1:t}) = \frac{1}{N}\sum_{i=1}^{N}\delta_{\mathbf{x}_t^{(i)}}(\mathrm{d}\mathbf{x}_t)$, and the estimate of $\pi_t(\varphi)$ in (6) is given by

$$\pi_t^N(\varphi) = \frac{1}{N}\sum_{i=1}^{N}\varphi(\mathbf{x}_t^{(i)}). \tag{7}$$

If the proposal is chosen as the transition density, i.e., $q_t(\mathbf{x}_t|\mathbf{x}_{1:t-1}^{(i)}, \mathbf{y}_t) = f_t(\mathbf{x}_t|\mathbf{x}_{t-1}^{(i)})$, we obtain the bootstrap particle filter (BPF) [4]. This corresponds to the simple procedure of sampling $\bar{\mathbf{x}}_t^{(i)}$ from $f_t(\mathbf{x}_t|\mathbf{x}_{t-1}^{(i)})$, and setting its weight $\mathsf{w}_t^{(i)} \propto g_t(\mathbf{y}_t|\bar{\mathbf{x}}_t^{(i)})$.

## 3 Generalised Bayesian filtering

### 3.1 A simple generalised particle filter

As explained in Section 2.1, given a standard probability model comprised of the prior $\pi_0(\mathbf{x})$ and a likelihood $g(\mathbf{y}|\mathbf{x})$, the general Bayes update defines an alternative, generalised likelihood $G(\mathbf{y}|\mathbf{x})$. The sequence of generalised likelihoods, denoted as $G_t(\mathbf{y}_t|\mathbf{x}_t)$ for $t \geq 1$, in an HMM yields a joint generalised posterior density which factorises as

$$p_t(\mathbf{x}_{0:t}|\mathbf{y}_{1:t}) \propto \pi_0(\mathbf{x}_0)\prod_{k=1}^{t}f_k(\mathbf{x}_k|\mathbf{x}_{k-1})G_k(\mathbf{y}_k|\mathbf{x}_k), \tag{8}$$

where $G_t(\mathbf{y}_t|\mathbf{x}_t) := \exp(-\ell_t(\mathbf{x}_t, \mathbf{y}_t))$. Inference can be done via SMC applied to this sequence of twisted probabilities defining a Feynman-Kac flow in the terminology of [32].

Comparing the update rule in (4) to the standard Bayes update suggests a generalisation of the particle filter. In particular, under the model in (1)–(3), one can perform generalised inference using $(f_t)_{t \geq 1}$ as usual, but replacing the likelihood with $(G_t)_{t \geq 1}$. Hence, a generalised sequential importance resampling PF (given fully in Algorithm 1) keeps the sampling step intact, but applies a different weight computation step $\mathsf{w}_t^{(i)} \propto \exp(-\ell(\bar{\mathbf{x}}_t^{(i)}, \mathbf{y}_t))\frac{f_t(\bar{\mathbf{x}}_t^{(i)}|\mathbf{x}_{t-1}^{(i)})}{q_t(\bar{\mathbf{x}}_t^{(i)}|\mathbf{x}_{1:t-1}^{(i)}, \mathbf{y}_t)}$. Indeed, most PFs (including the APF, see Algorithm 3 in the appendix) and related algorithms can be adapted to the GBI context.

**Algorithm 1** The generalised particle filter
---
**Input:** Observation sequence $\mathbf{y}_{1:T}$, number of samples $N$, proposal distributions $q_{1:T}(\cdot)$.
**Initialize:** Sample $\{\bar{\mathbf{x}}_0^{(i)}\}_{i=1}^N$ for the prior $\pi_0(\mathbf{x}_0)$.
**for** $t = 1$ **to** $T$ **do**
    **Sample:** $\bar{\mathbf{x}}_t^{(i)} \sim q_t(\mathbf{x}_t|\mathbf{x}_{1:t-1}^{(i)}, \mathbf{y}_t)$,     for   $i = 1, \ldots, N$.
    **Weight:** $\mathsf{w}_t^{(i)} \propto \exp(-\ell(\bar{\mathbf{x}}_t^{(i)}, \mathbf{y}_t)) \frac{f_t(\bar{\mathbf{x}}_t^{(i)}|\mathbf{x}_{t-1}^{(i)})}{q_t(\bar{\mathbf{x}}_t^{(i)}|\mathbf{x}_{1:t-1}^{(i)}, \mathbf{y}_t)}$,     for   $i = 1, \ldots, N$.
    **Resample:** $\mathbf{x}_t^{(i)} \sim \sum_{i=1}^N \mathsf{w}_t^{(i)} \delta_{\bar{\mathbf{x}}_t^{(i)}}(\mathrm{d}\mathbf{x}_t)$,     for   $i = 1, \ldots, N$.
**end for**
---

### 3.2 The $\beta$-BPF and the $\beta$-APF

The $\beta$-BPF is derived by selecting $\ell_t(\mathbf{x}_t, \mathbf{y}_t)$ as the $\beta$-divergence and applying the BPF procedure with the associated generalised likelihood. In this case, the loss is

$$\ell_t^\beta(\mathbf{x}_t, \mathbf{y}_t) = \frac{1}{\beta+1} \int g_t(\mathbf{y}_t'|\mathbf{x}_t)^{\beta+1} d\mathbf{y}_t' - \frac{1}{\beta} g_t(\mathbf{y}_t|\mathbf{x}_t)^\beta. \tag{9}$$

We can then construct the general $\beta$-likelihood as

$$G_t^\beta(\mathbf{y}_t|\mathbf{x}_t) \propto \exp(-\ell_t^\beta(\mathbf{x}_t, \mathbf{y}_t)). \tag{10}$$

In this instance, the use of the $\beta$-divergence provides the sampler with robust properties [11]. This can informally be seen from the form of the loss function in (9), where small values of $\beta$ temper the likelihood extending its tails making the loss more forgiving to outliers. The $\beta$-BPF procedure is given in Algorithm 2 in the appendix. The $\beta$-APF (Algorithm 3 in the appendix) is an Auxiliary Particle Filter [12, 33] adapted to the GBI setting, and is derived similarly to the $\beta$-BPF.

Note that the integral term in (9) is independent of $\mathbf{x}_t$ and can be absorbed, without evaluation, into the normalising constant when $\mathbf{x}_t$ is a location parameter for a symmetric $g_t(\cdot)$ and $\mathsf{Y}$ is a linear subspace of $\mathbb{R}^{d_y}$. More generally, if $g_t(\cdot)$ is a member of the exponential family, the integral can be computed by identifying $g_t^\beta(\cdot)$ with the kernel of another member of the same family with canonical parameters scaled by $\beta$. The overhead of computing $G_t^\beta(\cdot)$ is negligible in this instance, which is not too restrictive in the context of misspecitfied models. For other likelihoods, unbiased estimators for $G_t^\beta(\cdot)$, e.g. Poisson estimator [34], can be used in a random weight particle filter framework [35], where the overhead of computing $G_t^\beta(\cdot)$ will depend on the variance of the estimator and the convergence results from this setting apply but as [35] demonstrate this cost need not be prohibitive.

### 3.3 Selecting $\beta$

It is often the case that the primary goal of inference, particularly in the presence of model misspecification, is prediction. Hence, we propose choosing divergence parameters that lead to maximally predictive posterior belief distributions. In particular, for the $\beta$-BPF and $\beta$-APF, define $\mathcal{L}_\beta(\mathbf{y}_t, \hat{\mathbf{y}}_t)$ as a loss function of the observations $\mathbf{y}_t$ and the predictions $\hat{\mathbf{y}}_t$. We propose to choose $\beta$ as the solution to the following decision-theoretic optimisation problem:

$$\min_\beta \mathsf{agg}_{t=1}^T (\mathbb{E}_{p(\hat{\mathbf{y}}_t|\mathbf{y}_{1:t-1})} \mathcal{L}_\beta(\mathbf{y}_t, \hat{\mathbf{y}}_t)), \tag{11}$$

where agg denotes an aggregating function. This approach requires some training data to allow the selection of $\beta$. In filtering contexts, this can be historical data from the same setting or other available proxies. For offline inference one could also employ the actual data within this framework. Since, this proposal relies on the quality of the observations, which in the case of outlier contamination is violated by definition. To remedy this, we propose choosing robust versions for agg and $\mathcal{L}$, e.g. the median and the (standardised) absolute error respectively.

## 4 Theoretical guarantees

Theoretical guarantees for SMC methods can be extended to the generalised Bayesian filtering setting. Since the generalised Bayesian filters can be seen as a standard SMC methods with modified

likelihoods, the same analytical tools can be used in this setting. We provide guarantees for the $\beta$-BPF but emphasise that essentially the same results can be obtained much more broadly (including for the $\beta$-APF via the approach of [33]) using the standard arguments from the SMC literature.

We denote the generalised filters and generalised posteriors for the HMM in the $\beta$-divergence setting as $\pi_t^\beta$ and $\mathsf{p}_t^\beta$ respectively. Consequently, corresponding quantities constructed by the $\beta$-BPF are denoted as $\pi_t^{\beta,N}$ and $\mathsf{p}_t^{\beta,N}$. Although the generalised likelihoods $G_t^\beta(\mathbf{y}_t|\mathbf{x}_t)$ are not normalised, they can be considered as potential functions [32]. Since $G_t^\beta(\mathbf{y}_t|\mathbf{x}_t) < \infty$ whenever $g_t(\mathbf{y}_t|\mathbf{x}_t) < \infty$ and $\beta$ is fixed, we can adapt the standard convergence results into the generalised case.

**Assumption 1.** *For a fixed arbitrary observation sequence* $\mathbf{y}_{1:T} \in \mathsf{Y}^{\otimes T}$*, the potential functions* $(G_t^\beta)_{t \geq 1}$ *are bounded and* $G_t^\beta(\mathbf{y}_t|\mathbf{x}_t) > 0, \quad \forall t \in \{1, \ldots, T\}$ *and* $\mathbf{x}_t \in \mathsf{X}$.

This assumption holds for most used likelihood functions and their generalised extensions.

**Theorem 1.** *For any* $\varphi \in B(\mathsf{X})$ *and* $p \geq 1$,

$$\|\pi_t^{\beta,N}(\varphi) - \pi_t^\beta(\varphi)\|_p \leq \frac{c_{t,p,\beta}\|\varphi\|_\infty}{\sqrt{N}},$$

*where* $c_{t,p,\beta} < \infty$ *is a constant independent of* $N$.

The proof sketch and the constant $c_{t,p,\beta}$ are in the supplement. This $L_p$ bound provides a theoretical guarantee on the convergence of particle approximations to generalised posteriors. The special case when $p = 2$ also provides the error bound for the mean-squared error. It is well known that Theorem 1 with $p > 2$ leads to a law of large numbers via Markov's inequality and a Borel-Cantelli argument:

**Corollary 1.** *Under the setting of Theorem 1,* $\lim_{N \to \infty} \pi_t^{\beta,N}(\varphi) = \pi_t^\beta(\varphi)$ *a.s., for* $t \geq 1$.

Finally, a central limit theorem for estimates of expectations with respect to the smoothing distributions can be obtained by considering the path space $\mathsf{X}^{\otimes t}$. Recall the joint posterior $\mathsf{p}_t^\beta(\mathbf{x}_{1:t}|\mathbf{y}_{1:t})$ and consider a test function $\varphi_t : \mathsf{X}^{\otimes t} \to \mathbb{R}$. We denote $\overline{\varphi}_t^\beta := \int \varphi_t^\beta(\mathbf{x}_{1:t})\mathsf{p}_t^\beta(\mathbf{x}_{1:t}|\mathbf{y}_{1:t})$ and denote the $\beta$-BPF estimate of $\overline{\varphi}_t$ with $\overline{\varphi}_t^{\beta,N} := \int \varphi_t(\mathbf{x}_{1:t})\mathsf{p}_t^{\beta,N}(\mathrm{d}\mathbf{x}_{1:t})$.

**Theorem 2.** *Under the regularity conditions given in [36, Theorem 1],*

$$\sqrt{N}\left(\overline{\varphi}_t^{\beta,N} - \overline{\varphi}_t^\beta\right) \xrightarrow{d} \mathcal{N}\left(0, \sigma_{t,\beta}^2(\varphi_t)\right),$$

*as* $N \to \infty$ *where* $\sigma_{t,\beta}^2(\varphi_t) < \infty$.

The expression for $\sigma_{t,\beta}^2(\varphi_t)$ can be found in the appendix. These results illustrate that the standard guarantees for generic particle filtering methods extend to our case.

## 5   Experiments

In this section, we focus on $\beta$-BPF illustrating its the properties and empirically verifying its robustness. We include three experiments in the main text and another in Appendix E. Furthermore, we specifically investigate the $\beta$-APF in Section 5.2 comparing its behaviour to the $\beta$-BPF. Throughout, we report the *normalised mean squared error (NMSE)* and the *90% empirical coverage* as goodness-of-fit measures. The NMSE scores indicate the mean fit for the inferred posterior distribution and the empirical coverage measures the quality of its uncertainty quantification. We note that any claim in performance difference is based on the Wilcoxon signed-rank test. Further results and in-depth details on the experimental setup are given in the supplementary material.

### 5.1   A Linear-Gaussian state-space model

The Wiener velocity model [37] is a standard model in the target tracking literature, where the velocity of a particle is modelled as a Wiener process. The discretised version of this model can be represented as a Linear-Gaussian State-Space model (LGSSM),

$$\mathbf{x}_t = \mathbf{A}\mathbf{x}_{t-1} + \boldsymbol{\nu}_{t-1}, \quad \boldsymbol{\nu}_t \sim \mathcal{N}(\mathbf{0}, \mathbf{Q}), \quad (12) \qquad \mathbf{y}_t = \mathbf{H}\mathbf{x}_t + \boldsymbol{\epsilon}_t, \quad \boldsymbol{\epsilon}_t \sim \mathcal{N}(\mathbf{0}, \Sigma), \qquad (13)$$

where $\mathbf{A}, \mathbf{Q}$ are state-transition parameters dictated by the continuous-time model and $\mathbf{H}$ is the observation matrix (see Appendix). We simulate this model in two-dimensions with $\Sigma = \mathbf{I}$, contaminating the observations with a large scale, zero-mean Gaussian, $\mathcal{N}(0, 100^2)$ with probability $p_c$. Our aim is to obtain the filtering density under the heavily-contaminated setting where optimal filters struggle to perform. We compare the proposed $\beta$-BPF, for a range of values for $\beta$, to the standard BPF with a Gaussian likelihood (BPF), the (optimal) Kalman filter and an Oracle BPF with likelihood corresponding to the true generative model, i.e., with a Gaussian mixture likelihood with mixture components matching the noise processes and mixture probabilities matching contamination probability.

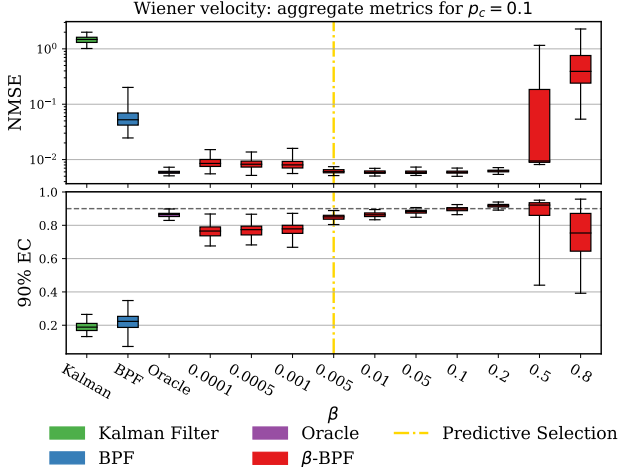

**Figure 1:** The mean metrics over state dimensions for the Wiener velocity example with $p_c = 0.1$. The top panel presents the NMSE results (lower is better) and the bottom panel presents the 90% empirical coverage results (higher is better), on 100 runs. The vertical dashed line in gold indicate the value of $\beta$ chosen by the selection criterion in Section 3.3. The horizontal dashed line in black in the lower panel indicates the 90% mark for the coverage.

We shed light onto four questions on this simple setup: (a) Does the $\beta$-BPF produce accurate and well-calibrated posterior distributions in the presence of contaminated data? (b) Is it sensitive to the choice $\beta$? (c) Does the method described in Section 3.3 for selecting $\beta$ return a near optimal result? (d) How does the robustification procedure compare to the inference with knowledge of the true model.

Figure 1 shows the results for $p_c = 0.1$. We observe that (a) the $\beta$-BPF outperforms the Kalman filter and the standard BPF for $\beta \leq 0.2$ while producing well-calibrated posteriors accounting for the uncertainty (for $\beta \in [0.01, 0.2]$ the coverage approaches the 90% threshold), (b) we see drastic performance gains (with median NMSE scores around $10\times$ smaller than the BPF and $100\times$ smaller that the Kalman filter) for a large range of $\beta$ values, (c) we also see that the $\beta$-choice heuristic [3] chooses a well-performing $\beta$ (gold vertical lines in Figure 1), and (d) that the performance of the $\beta$-BPF is very close the Oracle (with knowledge of the true model) for a range of $\beta$ values. Note that, for most values of $\beta$, the $\beta$-BPF significantly outperforms both the Kalman filter and the standard BPF predictively. The full set of results for the predictive performance are presented in Table 3 in Appendix G.1.

## 5.2 Terrain Aided Navigation

Terrain Aided Navigation (TAN) is a challenging estimation problem, where the state evolution is defined as in (12) (in three dimensions), but with a highly non-linear observation model, $\mathbf{y}_t = h(\mathbf{x}_t) + \boldsymbol{\epsilon}_t$, where $h(\cdot)$ is a non-linear function, typically including a non-analytic Digital Elevation Map (DEM). This problem simulates the trajectory of an aeroplane or a drone over a terrain map, where we observe its elevation over the terrain and its distance from its take-off hub from on-board sensors (see supplement for more details). We simulate transmission failure of the measurement system as impulsive noise on the observations, i.e., i.i.d. draws from a Student's $t$ distribution with $\nu = 1$ degrees of freedom. In other words, we define $\boldsymbol{\epsilon}_t \sim (1 - p_c)\mathcal{N}(0, 20^2) + p_c t_{\nu=1}(0, 20^2)$.

We apply both the $\beta$-BPF and the $\beta$-APF to this problem and compare them to the standard BPF with the Gaussian (BPF). We also compare against two other robust PF methods from the literature: Student's $t$ (t-BPF) [15] and the APF [12]. We set the degrees of freedom for the t-BPF to the same value as the contamination $\nu = 1$.

From Figure 2, we observe that for low contamination, both the $\beta$-BPF and the $\beta$-APF outperform the standard Gaussian BPF, the t-BPF and the APF. This shows that the use of $t$-distribution for the low contamination setting is inappropriate. This gap in the performance tightens, naturally, as $p_c$ grows since $t$-distribution becomes a good model for the observations. Notably, the performance gaps between the standard PFs and their $\beta$-robustified counterparts are similar, indicating that the use of the $\beta$-divergence in a particle filtering procedure does indeed robustify the inference.

In Figure 3, we plot the filtering distributions for the sixth state dimension (vertical velocity) obtained from an illustrative run with $p_c = 0.1$. The top panel shows the filtering distributions from the (Gaussian) BPF (up) and the $\beta$-BPF (down). The locations of the most prominent outliers are marked with dashed vertical lines in black. Figure 3 displays the significant difference between the two approaches: while the uncertainty for the standard BPF collapses when it meets the outliers, e.g. around $t = 1700$, the $\beta$-BPF does not suffer from this problem. This performance difference is partly related to the stability of the weights. The lower

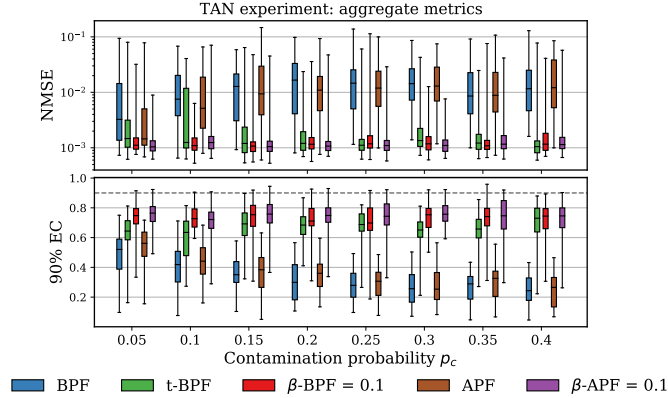

**Figure 2:** The mean metrics over state dimensions for the TAN example for different $p_c$. The top panel presents the NMSE results (lower is better) and the bottom panel presents the 90% empirical coverage results (higher is better), both evaluated on 50 runs. The horizontal dashed line in black in the lower panel indicate the 90% mark for the coverage.

panel in Figure 3 demonstrates the effective sample size (ESS) with time for the two filters showing that the $\beta$-BPF consistently exhibits larger ESS values, avoiding particle degeneracy. The ESS values for the BPF, on the other hand, sharply decline when it meets outliers. A similar result is observed for the APF versus the $\beta$-APF in the figures in the Appendix G.2. Further results on predictive performance can be found in Appendix G.2.

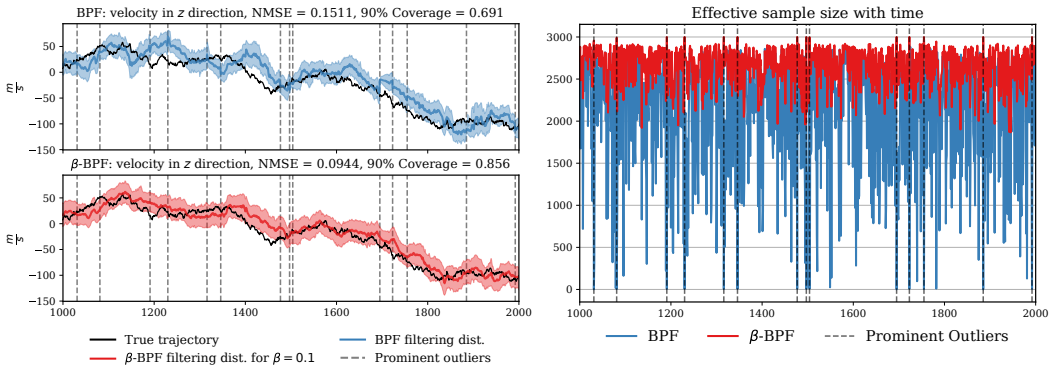

**Figure 3:** The left panel shows the inferred marginal filtering distributions for the velocity in the $z$ direction for the BPF and $\beta$-BPF with $\beta = 0.1$. The right panel shows the effective sample size with time. The locations of the most prominent (largest deviation) outliers are shown as dashed vertical lines in black in both panels.

## 5.3 London air quality Gaussian process regression

To measure air quality, London authorities use a network of sensors around the city recording pollutant measurements. Sensor measurements are susceptible to significant outliers due to environmental effects, manual calibration and sensor deterioration. In the experiment, we use Gaussian process (GP) regression to infer the underlying signal from a PM2.5 sensor.

For 1-D time series data, GP inference [38] can be accelerated to linear time in the number of observations by formulating an equivalent stochastic differential equation whose solution precisely

matches the GP under consideration [4] [23]. The resulting model is a LGSSM of the form (12)–(13) where the smoothing distribution matches the GP marginals at discrete-times. One can then apply smoothing algorithms, such as Rauch Tung Striebel (RTS) [39] or Forward Filters Backward Smoothing (FFBS) [40], to obtain the GP posterior. These require a forward filtering step with the Kalman filter for RTS or a PF for FFBS. Here, we fit a Matérn 5/2 GP with known hyperparameters to a time series from one of the sensors. We plot the median of the signals from the wider sensor network to obtain a simple approximation of the ground truth.

To further investigate the GP solution of the $\beta$-BPF (FFBS), we show the fit for $\beta = 0.1$ and compare it with Kalman (RTS) smoothing. In Figure 4 (and Figure 26 in the appendix) we see that the latter is sensitive to outliers forcing the GP mean towards them while the $\beta$-BPF is robust and ignores them.

Table 1 compares results with a Gaussian likelihood for GP regression with Kalman (RTS) smoothing, the standard BPF (FFBS) and two runs for the $\beta$-BPF (FFBS) ($\beta = 0.1$ by predictive selection as Section 3.3 and $\beta = 0.2$ by overall best performance). For both choices of $\beta$, the $\beta$-BPF outperforms all other methods on both metrics .

**Table 1:** GP regression NMSE (higher is better) and 90% empirical coverage for the credible intervals of the posterior predictive distribution, on 100 runs. **Bold** indicates statistically significant best result from Wilcoxon signed-rank test. All presented results are statistically different from each other according to the test.

| | median (IQR) | |
| --- | --- | --- |
| Filter (Smoother) | NMSE | EC |
| Kalman (RTS) | 0.144(0) | 0.685(0) |
| BPF (FFBS) | 0.116(0.015) | 0.650(0.020) |
| ($\beta = 0.1$)-BPF (FFBS) | 0.061(0.003) | 0.760(0.015) |
| ($\beta = 0.2$)-BPF (FFBS) | **0.059(0.002)** | **0.803(0.020)** |

## 6   Conclusions

We provided a generalised filtering framework based on GBI, which tackles likelihood misspecification in general state-space HMMs. Our approach leverages SMC methods, where we extended some analytical results to the generalised case. We presented the $\beta$-BPF, a simple instantiation of our approach based on the the $\beta$-divergence, developed an APF for this setting, and showed performance gains compared to other standard algorithms on a variety of problems and contamination settings[5].

This work opens up many exciting avenues for future research. Principle among which is online learning for model parameters (system identification) in the presence of misspecification. Our framework can directly incorporate most estimators found in the SMC literature and the computation of derivatives can be tackled with automatic differentiation tools.

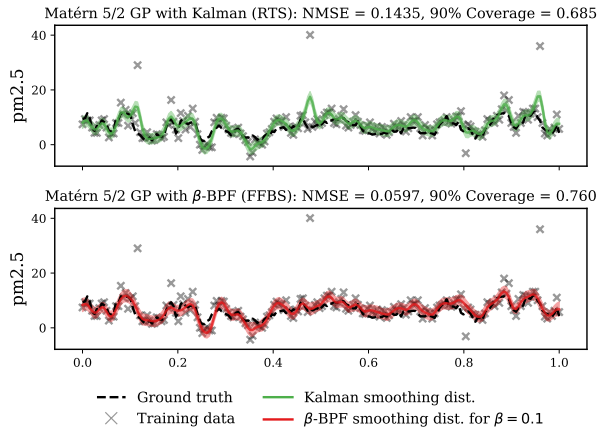

**Figure 4:** The GP fit on the measurement time series for one of the London air quality sensors. The top panel shows the posterior from the Kalman (RTS) smoothing. The bottom panel shows the posterior from the $\beta$-BPF (FFBS) for $\beta = 0.1$.

## Acknowledgements

This work was supported by the Lloyds Register Foundation programme on Data Centric Engineering through the London Air Quality project; The Alan Turing Institute for Data Science and AI under EPSRC grant EP/N510129/1; and the EPSRC under grants EP/R034710/1 and EP/T004134/1.

## Broader Impact

Robust inference in the context of misspecified models is a topic of broad interest. However, there are a few robust generally-applicable methods which can be employed in the context of online inference in time series settings. This paper provides a principled solution to this problem within a formal framework backed by theoretical guarantees and opening up the benefits to multiple application domains. The illustrative applications demonstrate the potential improvements in settings including navigation and Gaussian process regression, which, if realised more widely, could have wide-reaching impact. We hope that this inspires the community to build-on or apply our work to other challenging real-world scenarios.

Of particular interest is the application of Robust SMC methods, like the $\beta$-BPF and the auxiliary counterpart which were developed in this work, to impactful data-streaming applications in environmental monitoring and forecasting. Indeed, our research in this area was motivated by a real-world application in which existing techniques were inadequate (see https://www.turing.ac.uk/research/research-projects/london-air-quality for more details). We have demonstrated the benefits such methods in proof-of-concept work and are incorporating the resulting algorithms into a fully-developed platform, that has been in development for approximately three years. We are partnering with local authorities to help in directly informing policy makers and ultimately the general public.

More widely, this work provides an additional illustration that the GBI framework can provide good solutions to challenging problems in the world of misspecified framework and hence provides additional motivation to further investigate this extremely promising but rather new direction.

## Footnotes

[2]In the simplest form, drawing $N$ times with replacement from the weighted empirical measure to obtain an unweighted sample whose empirical distribution approximates the same target; see [31] for an overview of resampling schemes and their properties.

[3]We apply this choice criterion on an alternative dataset that is obtained from the same simulation but with 90% fewer observations.

[4]The SDE representation of a GP depends on the form of the covariance function. In this paper we use a GP with the Matérn 5/2 kernel, which admits a dual SDE representation.

[5]The code for this project is publicly available at `https://github.com/aboustati/robust-smc`.

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
