[Supplementary Material]

# Supplementary Material

## A  Generalized Bayesian Inference

Parametric Bayesian inference implicitly assumes that the generative model is well-specified, in particular, the observations are generated from the assumed likelihood model. In general, this assumption may not hold in real-world scenarios. Hence, one may wish to take into account the discrepancy between the true DGM and the assumed likelihood. Generalized Bayesian inference (GBI) is an approach proposed in [8] to deal with such cases.

For the simple Bayesian updating setup, consider a prior $\pi_0$ and the assumed likelihood function $g(\mathbf{y}|\mathbf{x})$. The posterior $\pi(\mathbf{x}|\mathbf{y}) =: \pi(\mathbf{x})$ is given by Bayes rule

$$\pi(\mathbf{x}) = \pi_0(\mathbf{x})\frac{g(\mathbf{y}|\mathbf{x})}{Z}, \tag{14}$$

where $Z := \int g(\mathbf{y}|\mathbf{x})\pi_0(\mathbf{x})\mathrm{d}\mathbf{x}$. [9] and [8] showed that (14) can be seen as a special case of a more general update rule, which can be described as a solution of an optimisation problem in the space of measures. In particular, let $L(\nu; \pi_0, \mathbf{y})$ be a loss-function where $\nu$ is a probability measure and $\pi_0$ is the prior, a belief distribution over $\mathbf{x}$ can be constructed by solving

$$\hat{\nu} = \arg\min_{\nu} L(\nu; \pi_0, \mathbf{y}). \tag{15}$$

To obtain Bayes-type updating rules, one needs to specify this loss function as a sum of a "data term" and a "regularisation term" [8] given as

$$L(\nu; \pi_0, \mathbf{y}) = \lambda_1(\nu, \mathbf{y}) + \lambda_2(\nu, \pi_0), \tag{16}$$

where $\lambda_1$ defines a data dependent "loss" and $\lambda_2$ controls the discrepancy between the prior and the final belief distribution $\hat{\nu}$. [8] show that the form of (16) that satisfies the von Neumann–Morgenstern utility theorem [41] and Bayesian additivity[6] is given by

$$L(\nu; \pi_0, \mathbf{y}) = \int \ell(\mathbf{x}, \mathbf{y})\nu(\mathrm{d}\mathbf{x}) + \mathrm{KL}(\nu||\pi_0), \tag{17}$$

which leads to a Bayes-type update [8, 42], given by

$$\pi(\mathbf{x}) = \pi_0(\mathbf{x})\frac{G(\mathbf{y}|\mathbf{x})}{Z}, \tag{18}$$

with $G(\mathbf{y}|\mathbf{x}) := \exp(-\ell(\mathbf{x}, \mathbf{y}))$ where $\ell(\mathbf{x}, \mathbf{y})$ is some divergence measuring the discrepancy between the observed information and the assumed model. In particular, if one assumes the real-world likelihood, i.e. the DGM, $h_0$, is different from the model likelihood $g$ and defines $\ell(\mathbf{x}, \mathbf{y})$ as a Kullback–Leibler (KL) divergence between the empirical likelihood $\tilde{h}_0$ (an empirical measure constructed using the observations) and the assumed likelihood $g(\mathbf{y}|\mathbf{x})$, the standard Bayes rule (14) arises as a solution. To see this, we can employ the KL divergence as a loss,

$$\mathrm{KL}(h_0||g) = \int \log h_0(\mathbf{y}')h_0(\mathrm{d}\mathbf{y}') - \int \log g(\mathbf{y}'|\mathbf{x})h_0(\mathrm{d}\mathbf{y}'),$$

and note that the first term does not affect the solution of the optimisation problem (15). Hence we arrive at the integrated loss function

$$\tilde{\ell}(\mathbf{x}) = -\int \log g(\mathbf{y}'|\mathbf{x})h_0(\mathrm{d}\mathbf{y}'). \tag{19}$$

By replacing the true likelihood $h_0$ with its empirical approximation upon observing $\mathbf{y}$, i.e., setting $h_0(\mathrm{d}\mathbf{y}') \approx \delta_{\mathbf{y}}(\mathrm{d}\mathbf{y}')$, we obtain $\tilde{\ell}(\mathbf{x}) \approx \ell(\mathbf{x}, \mathbf{y}) = -\log g(\mathbf{y}|\mathbf{x})$, which can be plugged in to (18) resulting in the standard Bayes update (14).

As previously mentioned, due to the properties of the KL divergence, the standard Bayes update is not robust to outliers [10]. Hence, substituting the KL with a more robust divergence such as the $\beta$-divergence, can endow inference with more robustness. Specifically, if $\ell$ is chosen as a $\beta$-divergence, the one step Bayes update for the likelihood $g(\mathbf{y}|\mathbf{x})$ can be written as

$$\pi(\mathbf{x}) = \pi_0(\mathbf{x}) \frac{G^\beta(\mathbf{y}|\mathbf{x})}{Z_\beta}, \tag{20}$$

where

$$G^\beta(\mathbf{y}|\mathbf{x}) = \exp\left(\frac{1}{\beta} g(\mathbf{y}|\mathbf{x})^\beta - \frac{1}{\beta+1} \int g(\mathbf{y}'|\mathbf{x})^{\beta+1} \mathrm{d}\mathbf{y}'\right). \tag{21}$$

One can then see $G^\beta(\mathbf{y}|\mathbf{x})$ as a generalised likelihood, resulting from the use of a different loss function compared to the standard Bayes procedure. Here $\beta$ is a hyperparameter that needs to be selected depending on the degree of misspecification. In general $\beta \in (0, 1)$ and $\lim_{\beta \to 0} G^\beta(\mathbf{y}|\mathbf{x}) = g(\mathbf{y}|\mathbf{x})$. Thus, intuitively, small $\beta$ values are suitable for mild model misspecification and large $\beta$ values are suitable when the assumed model is expected to significantly deviate from the true model.

## B  Influence Figure

The use of the $\beta$-divergence for updating the particle filter weights can be further motivated by studying the influence profile of the resulting weight update. Appendix B shows the influence that an observation exerts on the weights as a function of the number of standard deviations away from the mean. The figure compares the standard Gaussian likelihood, a Gaussian likelihood with an inflated variance, Student's t likelihood with 1 degree of freedom and a standard Gaussian warped by the $\beta$-divergence for 4 values of $\beta$. The plot shows that, with the $\beta$-divergence, observations that are close to the mean exert similar influence to the original standard Gaussian; however, the influence decreases away from the mean. This decrease is dependent on the value of $\beta$. For the case of an inflated Gaussian, the influence of the close observations is diminished compared to the original standard Gaussian; hence, this is not a suitable substitute to robustify the weight update since it deviates significantly from the properties of the assumed model near the mean. Finally, Student's t likelihood exerts higher influence on the inlying observations near the mean, which is also different from the assumed model.

## C  $\beta$-PF

### C.1  Outline derivation of the loss in (9)

To arrive at the experssion of the loss in (9), recall the formula for the beta divergence [11]

$$\begin{aligned} D_B^\beta(\mathbf{P}||\mathbf{Q}) =& \frac{1}{\beta(\beta+1)} \int (p^{\beta+1}(x) + \beta q^{\beta+1}(x) - (\beta+1)p(x)q^\beta(x)) \mathrm{d}\mu(x) \\ =& C_{\mathbf{P}} + \frac{1}{\beta+1} \int q^{\beta+1}(x) \mathrm{d}x - \frac{1}{\beta} \int q(x) \mathbf{P}(dx) \end{aligned}$$

where $\mathbf{P}$ and $\mathbf{Q}$ are probability measures on the measurable space $(X, \mathcal{A})$ and $\mu$ is a finite or $\sigma$-finite measure on this space, such that $\mathbf{P} \ll \mu$ and $\mathbf{Q} \ll \mu$ are absolutely continuous w.r.t. $\mu$ and $C_{\mathbf{P}}$ is a constant independent of $\mathbf{Q}$. Finally, $p = \frac{\mathrm{d}\mathbf{P}}{\mathrm{d}\mu}$ and $q = \frac{\mathrm{d}\mathbf{Q}}{\mathrm{d}\mu}$ are densities and the Radon-Nikodym derivatives for $\mathbf{P}$ and $\mathbf{Q}$ w.r.t. $\mu$.

Comparison with (17) yields (21) directly.

### C.2  $\beta$-BPF

Here, we provide the algorithmic procedure in Algorithm 2 for the $\beta$-BPF that is investigated in this main text.

**Figure 5:** This figure depicts the influence of a single observation on the particle weights for different likelihoods or generalised likelihoods.

---

**Algorithm 2** $\beta$-Bootstrap Particle Filter

---

**Input:** Observation sequence $\mathbf{y}_{1:T}$, number of samples $N$.
**Initialise:** Sample $\{\bar{\mathbf{x}}_0^{(i)}\}_{i=1}^N$ for the prior $\pi_0(\mathbf{x}_0)$.
**for** $t = 1$ **to** $T$ **do**
   **Sample:**
$$\tilde{\mathbf{x}}_t^{(i)} \sim f_t(\mathbf{x}_t|\bar{\mathbf{x}}_{t-1}^{(i)}) \quad \text{for } i = 1, \ldots, N.$$
   **Weight:**
$$\mathsf{w}_t^{(i)} \propto G_t^\beta(\tilde{\mathbf{x}}_t^{(i)}) \quad \text{for } i = 1, \ldots, N.$$
   **Resample:**
$$\bar{\mathbf{x}}_t^{(i)} \sim \sum_{i=1}^N \mathsf{w}_t^{(i)} \delta_{\tilde{\mathbf{x}}_t^{(i)}}(d\mathbf{x}_t) \quad \text{for } i = 1, \ldots, N.$$
**end for**

---

### C.3 $\beta$-APF

Here, we provide the algorithmic procedure in Algorithm 3 for the $\beta$-APF. Here $q_t$ denotes the proposal distribution at time $t$ which in the case of the fully-adapted APF would be chosen to be the conditional density of $\mathbf{x}_t$ given $\mathbf{x}_{t-1}$ and $\mathbf{y}_t$ but in general would be chosen as some approximation of this distribution and $\tilde{G}_t^\beta(\mathbf{x}_{t-1})$ is chosen as an approximation of the predictive generalised likelihood, i.e. $\tilde{G}_t^\beta(\mathbf{x}_{t-1}) \approx \int G_t^\beta(\mathbf{x}_t) f_t(\mathbf{x}_t|\mathbf{x}_{t-1}) d\mathbf{x}_t$.

As in the case of the standard APF, the use of reference points obtained from the current states in which one sets $\tilde{G}_t^\beta(x_{t-1}) = G_t^\beta(\mu_t(x_{t-1}))$ with $\mu_t(x_{t-1}) = \int x_t f(x_t|x_{t-1}) dx_t$ is one simple approach to this, but one which doesn't work well in full generality because it is underdispersed with respect to the true predictive generalised likelihood (cf. [33]). In general, better performance will

---

**Algorithm 3** $\beta$-Auxiliary Particle Filter

---

**Input:** Observation sequence $\mathbf{y}_{1:T}$, number of samples $N$.

**Initialise:** Sample $\{\bar{\mathbf{x}}_0^{(i)}\}_{i=1}^N$ independently from the prior $\pi_0(\mathbf{x}_0)$.

**for** $t = 1$ **to** $T$ **do**

   **Sample:**

$$k^{(i)} \sim \mathbb{P}\left(i = k|\mathbf{y}_t\right) \propto \mathsf{w}_{t-1}^{(i)}\tilde{G}_t^\beta(\bar{\mathbf{x}}_t^{(i)}) \qquad \text{for } i = 1, \ldots, N.$$

$$\bar{\mathbf{x}}_t^{(i)} \sim q_t(\bar{\mathbf{x}}_t|\bar{\mathbf{x}}_{t-1}^{k^{(i)}}) \qquad \text{for } i = 1, \ldots, N.$$

   **Weight:**

$$\mathsf{w}_t^{(i)} \propto \frac{f_t(\bar{\mathbf{x}}_t^{(i)}|\bar{\mathbf{x}}_{t-1}^{k^{(i)}})G_t^\beta(\bar{\mathbf{x}}_t^{(i)})}{q_t(\bar{\mathbf{x}}_t^{(i)}|\bar{\mathbf{x}}_{t-1}^{k^{(i)}})\tilde{G}_t^\beta(\bar{\mathbf{x}}_{t-1}^{k^{(i)}})} \quad \text{for } i = 1, \ldots, N.$$

  **end for**

---

of course be obtained by developing a good bespoke approximation to the predictive generalised likelihood and the locally-optimal proposal density for any given application, but in order to provide a simple generically-applicable strategy which is reasonably robust we suggest setting the proposal equal to the transition density, $q_t = f_t$, and using a stabilised version of the simple approximation to the predictive likelihood, provided by

$$\tilde{G}_t^\beta(x_{t-1}) = G_t^\beta(\mu_t(x_{t-1})) + c_t$$

where $c_t$ is a constant chosen, e.g. as $0.05 \sup_x G_t^\beta(x)$ to avoid any instability in the weighting step. Such a strategy was advocated in the iterated version of this algorithm described by [43] which could in principle also be adapted to the GBI setting.

# D  Theoretical analysis

## D.1  Proof of Theorem 1

This is an adaptation of a well-known proof, hence we will sketch results and provide the constant $c_{t,p,\beta}$.

The result is proved via induction. For $t = 0$, we have the result in the theorem trivially, as it corresponds to the i.i.d. case and, e.g. [32, Lemma 7.3.3] provides an explicit constant. Hence, as an induction hypothesis, we assume

$$\|\pi_{t-1}^{\beta,N}(\varphi) - \pi_{t-1}^{\beta}(\varphi)\|_p \leq \frac{c_{t-1,p,\beta}\|\varphi\|_\infty}{\sqrt{N}}, \tag{22}$$

where $c_{t-1,p,\beta} < \infty$ is independent of $N$. After the sampling step, we obtain the predictive particles $\bar{\mathbf{x}}_t^{(i)}$ and form the predictive measure

$$\bar{\pi}_t^{\beta,N}(\mathrm{d}\mathbf{x}_t|y_{1:t-1}) = \frac{1}{N}\sum_{i=1}^N \delta_{\bar{x}_t^{(i)}}(\mathrm{d}\mathbf{x}_t),$$

and then one can show that we have [44, Lemma 1]

$$\|\bar{\pi}_t^{\beta,N}(\varphi) - \bar{\pi}_t^{\beta}(\varphi)\|_p \leq \frac{c_{1,t,p,\beta}\|\varphi\|_\infty}{\sqrt{N}}, \tag{23}$$

where $c_{1,t,p,\beta} < \infty$ is a constant independent of $N$. After the computation of weights, we construct

$$\widetilde{\pi}_t^{\beta,N}(\mathrm{d}\mathbf{x}_t) = \sum_{i=1}^N \mathsf{w}_t^{(i)}\delta_{\widetilde{\mathbf{x}}_t^{(i)}}(\mathrm{d}\mathbf{x}_t). \tag{24}$$

Following again [44, Lemma 1], one readily obtains

$$\|\pi_t^{\beta}(\varphi) - \widetilde{\pi}_t^{\beta,N}(\varphi)\|_p \leq \frac{c_{2,t,p,\beta}\|\varphi\|_\infty}{\sqrt{N}}, \tag{25}$$

where

$$c_{2,t,p,\beta} = \frac{2\|G_t^{\beta}\|_\infty c_{1,t,p,\beta}}{\bar{\pi}_t(G_t^{\beta})} < \infty,$$

where we note $\bar{\pi}_t(G_t^{\beta}) > 0$. Finally, performing multinomial resampling leads to a conditionally-i.i.d. sampling case, which yields

$$\|\widetilde{\pi}_t^{\beta,N}(\varphi) - \pi_t^{\beta,N}(\varphi)\|_p \leq \frac{c_{3,t,p,\beta}\|\varphi\|_\infty}{\sqrt{N}}. \tag{26}$$

Combining (25) and (26) yields the result with $c_{t,p,\beta} = c_{2,t,p,\beta} + c_{3,t,p,\beta}$.

## D.2  Proof of Corollary 1

We sketch here a standard argument for obtaining a strong law of large numbers from $L_p$ error bounds. Let us write for simplicity that

$$\xi_N = \pi_t^{\beta,N}(\varphi) \qquad \text{and} \qquad \xi = \pi_t^{\beta}(\varphi). \tag{27}$$

The strategy is to note that

$$\left\{\lim_{k\to\infty}|\xi_k - \xi| = 0\right\} = \bigcap_{l=1}^\infty \left\{\lim_{k\to\infty}|\xi_k - \xi| < 1/l\right\}$$

and hence if it can be shown that the $\mathbb{P}(\{|\xi_k - \xi| < 1/l\}) \to 1$ for every $l \in \mathbb{N}$ then the event that $\xi_k \to \xi$ is a countable intersection of events of probability 1 and hence itself has probability 1.

Using the Borel-Cantelli lemma (see, e.g. [45, p. 255]), to show that $\mathbb{P}(|\xi_k - \xi| \geq \varepsilon) \to 0$ as $k \to \infty$ it suffices to demonstrate that

$$\sum_{k=1}^{\infty} \mathbb{P}(|\xi_k - \xi| \geq \varepsilon) < \infty.$$

We do this via the generalised Markov's inequality:

$$\mathbb{P}(|\xi_k - \xi| \geq \varepsilon) \leq \frac{\mathbb{E}[|\xi_k - \xi|^p]}{\varepsilon^p},$$

which combined with Theorem 1 yields

$$\mathbb{P}(|\xi_k - \xi| \geq \varepsilon) \leq \frac{c^p \|\varphi\|_\infty^p}{k^{p/2} \varepsilon^p}.$$

Choosing any $p > 2$ ensures that the rhs is summable and hence that $\mathbb{P}(|\xi_k - \xi| < \varepsilon) \to 1$ as $k \to \infty$ for any $\varepsilon > 0$ and, by taking $\varepsilon = 1/l$ for each $l \in \mathbb{N}$, the proof is complete.

### D.3   Proof of Theorem 2

We refer to the Proposition in [33] which provides explicit expressions for sequential importance resampling based particle filters within the general frameworks of [32, 36]; the same argument holds *mutatis mutandis* in the context of the $\beta$-BPF. We note that the asymptotic variance expression $\sigma_{t,\beta}^2(\varphi)$ is given as follows. For $t = 1$, we obtain [33]

$$\sigma_{1,\beta}^2(\varphi) = \int \frac{\mathsf{p}_1^\beta(\mathbf{x}_1|\mathbf{y}_1)}{f_1(\mathbf{x}_1)} (\varphi_1(\mathbf{x}_1) - \overline{\varphi}_1)^2 \mathrm{d}\mathbf{x}_1,$$

where $f_1(\mathbf{x}_1) := \int \mu_0(\mathbf{x}_0) f_1(\mathbf{x}_1|\mathbf{x}_0) \mathrm{d}\mathbf{x}_0$. Then, for $t > 1$ [33]

$$\sigma_{t,\beta}^2 = \int \frac{\mathsf{p}_t^\beta(\mathbf{x}_1|\mathbf{y}_{1:t})^2}{f_1(\mathbf{x}_1)} \left( \int \varphi_t(\mathbf{x}_{1:t}) \mathsf{p}_t^\beta(\mathbf{x}_{2:t}|\mathbf{y}_{2:t}, \mathbf{x}_1) \mathrm{d}\mathbf{x}_{2:t} - \overline{\varphi}_t \right)^2 \mathrm{d}\mathbf{x}_1$$

$$+ \sum_{k=2}^{t-1} \int \frac{\mathsf{p}_k^\beta(\mathbf{x}_{1:k}|\mathbf{y}_{1:t})^2}{\mathsf{p}_{k-1}^\beta(\mathbf{x}_{1:k-1}|\mathbf{y}_{1:k-1}) f_k(\mathbf{x}_k|\mathbf{x}_{k-1})} \left( \int \varphi_t(\mathbf{x}_{1:t}) \mathsf{p}_t^\beta(\mathbf{x}_{k+1:t}|\mathbf{y}_{k+1:t}, \mathbf{x}_k) \mathrm{d}\mathbf{x}_{k+1:t} - \overline{\varphi}_t \right)^2 \mathrm{d}\mathbf{x}_{1:k}$$

$$+ \int \frac{\mathsf{p}_t^\beta(\mathbf{x}_{1:t}|\mathbf{y}_{1:t})^2}{\mathsf{p}_{t-1}^\beta(\mathbf{x}_{1:t-1}|\mathbf{y}_{1:t-1}) f_t(\mathbf{x}_t|\mathbf{x}_{t-1})} (\varphi_t(\mathbf{x}_{1:t}) - \overline{\varphi}_t)^2 \mathrm{d}\mathbf{x}_{1:t}.$$

## E   Asymmetric Wiener velocity

In the case of simple, symmetric noise settings with additive contamination the use of heavy-tailed likelihoods such as Student's $t$ may be still seen as a viable alternative to robustify the inference. However, there are some realistic settings in which such off-the-shelf heavy-tailed replacements are not feasible or require considerable model-specific work. Consider, as a simple illustration, the Wiener velocity example in Section 5.1, where the observation noise in (13) is replaced with $\epsilon_t \sim \mathbb{1}_{[-\infty,0]} \mathcal{N}(0,1) + \mathbb{1}_{[0,+\infty]} \mathcal{N}(0,10^2)$. This simulates an asymmetric noise scenario. The observations are further contaminated with multiplicative exponential noise, i.e. $\epsilon_t \leftarrow \xi \epsilon_t$, for $\xi \sim \text{Exp}(1000)$ with probability $p_c$. This sums up to a multiplicatively corrupted asymmetric noise distribution which could, for example, represent a sensor with asymmetric noise profile in a failing regime which occasionally exhibits excessive gain.

For this example, it is easy to derive a BPF with the assymetric likelihood. It is also easy to extend this likelihood to the $\beta$-BPF case. We test BPF and the $\beta$-BPF ($\beta = 0.1$) versus two versions of the t-BPF, in which the likelihood is replaced with a heavy-tailed symmetric one, one set to a short scale $\sigma = 1$ and the other set to a long scale $\sigma = 10$.

Figure 6 shows the results for this experiment. The BPF is unable to handle the multiplicative exponential contamination, as can be seen by the NMSE values. It also provides poor posterior coverage. The t-BPF fairs better with this type of contamination where we can see a trade-off between

b

Asymmetric Wiener velocity: aggregate metrics for $p_c = 0.1$

**Figure 6:** The mean metrics over state dimensions for the asymmetric Weiner velocity example with $p_c = 0.1$. The left panel presents the NMSE results (lower is better) and the right panel presents the 90% empirical coverage results (higher is better), evaluated on 100 runs. The $x$-axis ticks indicate the scale used for Student's $t$ likelihood. The horizontal dashed line in black in the right panel indicates the 90% mark for the coverage.

accuracy and coverage depending on the chosen scale of the likelihood. This is due to the symmetry of the $t$-distribution which overestimates one of the tails depending on the scale. The $\beta$-BPF does not have this trade-off and outperforms the t-BPF on both metrics.

While one might attempt to model the noise with an asymmetric construction of the $t$-distributions which approximates the noise structure, we argue that in more general settings using heavy-tailed distributions requires approximations of the noise structure and making modelling choices which could be arbitrarily complex. This is in contrast to specifying a single tuning parameter as in the $\beta$-divergence case. The $\beta$-BPF requires no further modelling than the original problem and can be used as a drop-in replacement for nearly all types of likelihood structures.

## F  Experiment Details

### F.1  Evaluation Metrics

The following metrics metrics are used to evaluate the experiments:

**The Normalised Mean Squared Error (NMSE)**   is computed per state dimension $j$ as

$$\text{NMSE}_j = \frac{\left\|\sum_{t=1}^{T} x_{tj} - \hat{x}_{tj}\right\|_2^2}{\sum_{t=1}^{T} \|x_{tj}\|_2^2},$$

(28)

with $\hat{x}_{tj} = \frac{1}{N} \sum_{i=1}^{N} \bar{x}_{tj}^{(i)}$, i.e. the mean over resampled particles (trajectories).

**The 90% Emprical Coverage (EC)**   is computed per state dimension $j$ as

$$\text{EC}_j = \frac{\sum_{t=1}^{T} \mathbb{1}_{C_t}(x_{tj})}{T},$$

(29)

with

$$C_t = \{z | z \in [q_{0.05}(\{\bar{x}_{tj}^{(i)}\}_{i=1}^{N}), q_{0.95}(\{\bar{x}_{tj}^{(i)}\}_{i=1}^{N})]\},$$

where q is the quantile function.

**Predictive Median Absolute Error (MedAE)**   is computed per observation dimension $j$ as

$$\text{MedAE} = \text{MEDIAN}_{t \in \{1, \ldots, T\}} \left(|\hat{y}_{tj} - y_{tj}|\right),$$

(30)

where $\hat{y}_t \sim \sum_{i=1}^{N} w_t^i g_t(\mathbf{y}|\mathbf{x}_t^{(i)})$.

**Aggregation:** Metrics are often presented as aggregates over the state dimensions, which are simply the mean of the metric across the state dimensions.

## F.2 Details on the implementation of the selection criterion in Section 3.3

From (11), we chose agg as the median and $\mathcal{L}$ as the absolute error. When the observations are multidimensional, we take the average loss weighted by the inverse of the median of each dimension.

We compute the score for different values of $\beta$ from a grid and choose $\beta$ that minimises the score. For multiple runs, we report the modal value of the $\beta$'s over all the runs.

In the interest of simplicity, we use the entire observation sequence from an alternative realisation of the same simulation to compute the score. However, in practice one might one to tune $\beta$ on a sub-sequence to avoid extra computation.

## F.3 Wiener velocity model experiment details (Section 5.1)

In this section, we detail the experimental setup used to obtain the results for Section 5.1.

**Simulator settings** We synthesise the data with a Python simulator utilising NumPy. We discretise the system with $\Delta\tau = 0.1$ and simulate it for 100 time steps, i.e. we obtain 1000 time points in total. For the state evolution process in Equation (12), we set the transition matrix $\mathbf{A} = \begin{bmatrix} 1 & 0 & \Delta\tau & 0 \\ 0 & 1 & 0 & \Delta\tau \\ 0 & 0 & 1 & 0 \\ 0 & 0 & 0 & 1 \end{bmatrix}$

and the transition covariance matrix $\mathbf{Q} = \begin{bmatrix} \frac{\Delta\tau^3}{3} & 0 & \frac{\Delta\tau^2}{2} & 0 \\ 0 & \frac{\Delta\tau^3}{3} & 0 & \frac{\Delta\tau^2}{2} \\ \frac{\Delta\tau^2}{2} & 0 & \Delta\tau & 0 \\ 0 & \frac{\Delta\tau^2}{2} & 0 & \Delta\tau \end{bmatrix}$. For the observation process in

Equation (13), we set the observation matrix $\mathbf{H} = \begin{bmatrix} 1 & 0 & 0 & 0 \\ 0 & 1 & 0 & 0 \end{bmatrix}$ and the noise covariance $\Sigma = \mathbf{I}$. The initial state of the simulator is set to $\mathbf{x}_0 = [140, 140, 50, 0]$.

**Contamination** To simulate contaminated observations we apply extra i.i.d. Gaussian noise with a standard deviation of 100.0 to the observation sequence with probability $p_c$ per observation.

**Sampler settings** We initialise the samplers by sampling from the prior given by $\mathcal{N}(\mathbf{x}_0, \mathbf{Q})$ with $\mathbf{x}_0$ being the initial state of the simulator and $\mathbf{Q}$ as above. We set the likelihood covariance to the simulator noise covariance and the number of samples to 1000.

**Experiment settings** Each experiment consists of 100 runs, where all samplers are seeded with the same seed per run; however, the seeds vary across the runs. We use the same state sequence for all runs obtained from the simulator as above. However, each run simulates a new observation sequence (i.e. the observations noise changes per run).

## F.4 Terrain Aided Navigation (TAN) experiment details (Section 5.2)

In this section, we detail the experimental setup used to obtain the results for Section 5.2.

**Simulator settings** We synthesise the data with a Python simulator utilising NumPy. We discretise the system with $\Delta\tau = 0.1$ and simulate it for 200 time steps, i.e. we obtain 2000 time points in total. For the state evolution process in Equation (12), we set the transition matrix

$$\mathbf{A} = \begin{bmatrix} 1 & 0 & 0 & \Delta\tau & 0 & 0 \\ 0 & 1 & 0 & 0 & \Delta\tau & 0 \\ 0 & 0 & 1 & 0 & 0 & \Delta\tau \\ 0 & 0 & 0 & 1 & 0 & 0 \\ 0 & 0 & 0 & 0 & 1 & 0 \\ 0 & 0 & 0 & 0 & 0 & 1 \end{bmatrix},$$

and the transition covariance matrix

$$\mathbf{Q} = \begin{bmatrix} 4 & 0 & 0 & 0 & 0 & 0 \\ 0 & 4 & 0 & 0 & 0 & 0 \\ 0 & 0 & 36 & 0 & 0 & 0 \\ 0 & 0 & 0 & 0.0841 & 0 & 0 \\ 0 & 0 & 0 & 0 & 0.207936 & 0 \\ 0 & 0 & 0 & 0 & 0 & 5.29 \end{bmatrix}.$$

For the observation process, we set the non-linear observation function

$$h(\mathbf{x}_t) = \begin{bmatrix} x_{t3} - \text{DEM}(x_{t1}, x_{t2}) \\ \sqrt{(x_{t1} - x_{01})^2 + (x_{t2} - x_{02})^2} \end{bmatrix},$$

where DEM is a non-analytic Digital Elevation Map. For our simulation we set DEM to

$$\text{DEM(a, b)} = \text{peaks}(q \cdot a, q \cdot b) + \sum_{i=1}^{6} \alpha_i \sin(\omega_i \cdot q \cdot a) \cos(\psi \cdot q \cdot b),$$

with $\text{peaks}(c, d) = 200(3(1-c)^2 \exp(-c^2 - (d+1)^2) - 10(\frac{c}{5} - c^3 - d^5) \exp(-c^2 - d^2) - \frac{1}{3} \exp(-(x+1)^2 + y^2))$, $\boldsymbol{\alpha} = [300, 80, 60, 40, 20, 10]$, $\boldsymbol{\omega} = [5, 10, 20, 30, 80, 150]$, $\boldsymbol{\psi} = [4, 10, 20, 40, 90, 150]$ and $q = \frac{3}{2.96 \times 10^4}$. The noise covariance $\Sigma = \sigma^2 \mathbf{I}$ with $\sigma^2 = 400$. The initial state of the simulator is set $\mathbf{x}_0 = [-7.5 \times 10^3, 5 \times 10^3, 1.1 \times 10^3, 88.15, -60.53, 0]$.

**Contamination**  To simulate contaminated observations we apply extra i.i.d. Student's t noise with 1 degree of freedom scale $\sigma$, where $\sigma$ is given as above. The contamination is applied to observation instances with probability $p_c$ per observation.

**Sampler settings**  We initialise the samplers by sampling from the prior given by $\mathcal{N}(\mathbf{x}_0, \mathbf{Q})$ with $\mathbf{x}_0$ being the initial state of the simulator and $\mathbf{Q}$ as above. We set the likelihood covariance to the simulator noise covariance and the number of samples to 3000. For the APFs, we make the same design choices outlined in Appendix C.3, i.e. setting the proposal density to the transition density and stabilising the predictive likelihood approximation with the given additive constant.

**Experiment settings**  Each experiment consists of 50 runs, where all samplers are seeded with the same seed per run; however, the seeds vary across the runs. We use the same state sequence for all runs obtained from the simulator as above. However, each run simulates a new observation sequence (i.e. the observation noise changes per run).

## F.5 Asymmetric Wiener velocity model experiment details (Appendix E)

In this section, we detail the experimental setup used to obtain the results for Appendix E.

**Simulator settings**  We use the same simulator settings as in Appendix F.3, but changing the observation noise to $\mathbb{1}_{[-\infty,0]}\mathcal{N}(0, 1) + \mathbb{1}_{[0,+\infty]}\mathcal{N}(0, 10^2)$.

**Contamination**  To simulate contaminated observations we multiplicative apply i.i.d. Exponential noise with a scale of 1000 with probability $p_c = 0.1$ per observation.

**Sampler settings**  We initialise the samplers by sampling from the prior given by $\mathcal{N}(\mathbf{x}_0, \mathbf{Q})$ with $\mathbf{x}_0$ being the initial state of the simulator and $\mathbf{Q}$ as above. We set the number of samples to 1000.

**Experiment settings**  We use the same settings as in Appendix F.3.

## F.6 Air quality experiment details (Section 5.3)

In this section, we detail the setup used to obtain the results for Section 5.3.

**Data**  The data was obtained from https://www.londonair.org.uk/london/asp/datadownload.asp. We select a time window of 200 hours. No preprocessing was performed on the data.

**Kernel** We use the Matérn 5/2 kernel and set the lengthscale $l = 0.03$ and the signal variance $\sigma_s^2 = 32$. We discretize the SDE representation of the Matérn 5/2 GP with stepsize $\Delta\tau = 0.005$ to obtain an LGSSM of the form (12)-(13), with transition matrix

$$\mathbf{A} = \exp(\Delta\tau\mathbf{F}) = \exp\left(\Delta\tau \begin{bmatrix} 0 & 1 & 0 \\ 0 & 0 & 1 \\ -\lambda^3 & -3\lambda^2 & -3\lambda \end{bmatrix}\right),$$

where $\lambda = \frac{\sqrt{5}}{l}$ and transition covariance matrix $\mathbf{Q} = \mathbf{P}_\infty - \mathbf{A}\mathbf{P}_\infty\mathbf{A}^\mathsf{T}$, with $\mathbf{P}_\infty = \begin{bmatrix} \sigma_s^2 & 0 & \kappa \\ 0 & \kappa & 0 \\ -\kappa & 0 & \sigma_s^2\lambda^4 \end{bmatrix}$,

where $\kappa = \frac{\sigma_s^2\lambda^2}{3}$. For the observation process in (13), the observation matrix is set to $\mathbf{H} = [1, 0, 0]$ and the noise variace $\sigma^2 = 1$. The prior on the initial state $\mathbf{x}_x$ is given as $\mathcal{N}(m, S)$, where $m^\mathsf{T} = [0, 0, 0]$ and $S = \mathbf{P}_\infty$.

**Sampler settings** We initialise the samples by sampling from the prior $\mathcal{N}(m, S)$. We set the number of samples to 1000.

**Smoother settings** We set the number of samples to 1000 for the FFBS smoother.

**Experiment settings** We repeat the sampling procedure for 100 runs, where the samplers are seeded differently for each runs. The seeds are shared among samplers for each run. The Kalman filter does not require multiple runs as the solution is deterministic.

# G Further results

## G.1 Wiener velocity experiment

Wiener velocity: aggregate metrics for $p_c = 0.0$

Wiener velocity: aggregate metrics for $p_c = 0.05$

Wiener velocity: aggregate metrics for $p_c = 0.1$

Wiener velocity: aggregate metrics for $p_c = 0.15$

Wiener velocity: aggregate metrics for $p_c = 0.2$

Wiener velocity: aggregate metrics for $p_c = 0.25$

Wiener velocity: aggregate metrics for $p_c = 0.3$

**Figure 7:** The mean metrics over state dimensions for the Wiener velocity example. The top panel presents the NMSE results (lower is better) and the bottom panel presents the 90% emprirical coverage results (higher is better), on 100 runs. The vertical dashed line in gold indicate the value of $\beta$ chosen by the selection criterion in Section 3.3. The horizontal dashed line in black in the lower panel indicates the 90% mark for the coverage.

**Figure 8:** Marginal filtering distributions for the Kalman filter, the BPF and the $\beta$-BPF.

**Figure 9:** Marginal filtering distributions for the Kalman filter, the BPF and the $\beta$-BPF.

**Figure 10:** Marginal filtering distributions for the Kalman filter, the BPF and the $\beta$-BPF.

**Figure 11:** Marginal filtering distributions for the Kalman filter, the BPF and the $\beta$-BPF.

|                    | Predictive Median Absolute Error | |
| Filter             | mean   | standard error |
| ------------------ | ------ | -------------- |
| Kalman Filter      | 5.23   | 0.06           |
| BPF                | 2.78   | 0.09           |
| $\beta = 0.0001$   | 0.97   | 0.00           |
| $\beta = 0.0005$   | 0.97   | 0.00           |
| $\beta = 0.001$    | 0.97   | 0.00           |
| $\beta = 0.005$    | 0.90   | 0.00           |
| $\beta = 0.01$     | 0.90   | 0.00           |
| $\beta = 0.05$     | 0.90   | 0.00           |
| $\beta = 0.1$      | 0.90   | 0.00           |
| $\beta = 0.2$      | 0.92   | 0.00           |
| $\beta = 0.5$      | 72.22  | 12.34          |
| $\beta = 0.8$      | 226.61 | 11.62          |

**Table 2:** Predictive results on the Weiner velocity example for $p_c = 0.1$. The one step ahead predictive performance is measure by the median absolute error. The figures are averaged across 100 runs and the standard error on the average score is provided.

## G.2  TAN experiment

**Figure 12:** Marginal filtering distributions for the BPF (top) and $\beta$-BPF (bottom) with $\beta = 0.1$. The locations of the most prominent (largest deviation) outliers are shown as dashed vertical lines in black.

**Figure 13:** Marginal filtering distributions for the BPF (top) and $\beta$-BPF (bottom) with $\beta = 0.1$. The locations of the most prominent (largest deviation) outliers are shown as dashed vertical lines in black.

**Figure 14:** Marginal filtering distributions for the BPF (top) and $\beta$-BPF (bottom) with $\beta = 0.1$. The locations of the most prominent (largest deviation) outliers are shown as dashed vertical lines in black.

**Figure 15:** Marginal filtering distributions for the BPF (top) and $\beta$-BPF (bottom) with $\beta = 0.1$. The locations of the most prominent (largest deviation) outliers are shown as dashed vertical lines in black.

**Figure 16:** Marginal filtering distributions for the BPF (top) and $\beta$-BPF (bottom) with $\beta = 0.1$. The locations of the most prominent (largest deviation) outliers are shown as dashed vertical lines in black.

**Figure 17:** Marginal filtering distributions for the BPF (top) and $\beta$-BPF (bottom) with $\beta = 0.1$. The locations of the most prominent (largest deviation) outliers are shown as dashed vertical lines in black.

**Figure 18:** Effective sample size with time for the BPF (top) and $\beta$-BPF with $\beta = 0.1$.

**Figure 19:** Marginal filtering distributions for the APF (top) and $\beta$-APF (bottom) with $\beta = 0.1$. The locations of the most prominent (largest deviation) outliers are shown as dashed vertical lines in black.

**Figure 20:** Marginal filtering distributions for the APF (top) and $\beta$-APF (bottom) with $\beta = 0.1$. The locations of the most prominent (largest deviation) outliers are shown as dashed vertical lines in black.

**Figure 21:** Marginal filtering distributions for the APF (top) and $\beta$-APF (bottom) with $\beta = 0.1$. The locations of the most prominent (largest deviation) outliers are shown as dashed vertical lines in black.

**Figure 22:** Marginal filtering distributions for the APF (top) and $\beta$-APF (bottom) with $\beta = 0.1$. The locations of the most prominent (largest deviation) outliers are shown as dashed vertical lines in black.

**Figure 23:** Marginal filtering distributions for the APF (top) and $\beta$-APF (bottom) with $\beta = 0.1$. The locations of the most prominent (largest deviation) outliers are shown as dashed vertical lines in black.

**Figure 24:** Marginal filtering distributions for the APF (top) and $\beta$-APF (bottom) with $\beta = 0.1$. The locations of the most prominent (largest deviation) outliers are shown as dashed vertical lines in black.

**Figure 25:** Effective sample size with time for the APF (top) and $\beta$-APF with $\beta = 0.1$.

| Filter | $p_c$ | | | | | | | |
|---|---|---|---|---|---|---|---|---|
| | 0.05 | 0.1 | 0.15 | 0.2 | 0.25 | 0.3 | 0.35 | 0.4 |
| BPF | 16.63(0.06) | 17.67(0.05) | 17.88(0.06) | 18.66(0.07) | 19.68(0.08) | 20.12(0.09) | 20.96(0.08) | 21.55(0.09) |
| t-BPF | 16.33(0.05) | 17.15(0.05) | 17.15(0.05) | 17.92(0.05) | 18.72(0.06) | 18.95(0.07) | 19.71(0.09) | 20.11(0.08) |
| $\beta$-BPF = 0.005 | 16.26(0.05) | 17.01(0.05) | 16.96(0.06) | 17.60(0.05) | 18.34(0.07) | 18.48(0.06) | 19.24(0.06) | 19.60(0.07) |
| $\beta$-BPF = 0.01 | 16.23(0.04) | 16.91(0.05) | 16.65(0.05) | 17.06(0.05) | 17.74(0.05) | 17.86(0.06) | 18.43(0.05) | 18.61(0.06) |
| $\beta$-BPF = 0.05 | 16.39(0.04) | 16.97(0.05) | 16.70(0.06) | 17.23(0.06) | 18.03(0.06) | 17.84(0.06) | 18.45(0.07) | 18.78(0.08) |
| $\beta$-BPF = 0.1 | 17.46(0.05) | 17.92(0.06) | 17.90(0.11) | 18.61(0.12) | 19.49(0.11) | 19.15(0.10) | 19.76(0.10) | 20.24(0.11) |
| $\beta$-BPF = 0.2 | 16.56(0.04) | 17.07(0.05) | 16.58(0.04) | 17.43(0.04) | 17.87(0.06) | 17.85(0.05) | 18.56(0.06) | 18.84(0.06) |
| APF | 15.96(0.04) | 17.09(0.04) | 17.34(0.05) | 18.13(0.05) | 19.04(0.08) | 19.51(0.06) | 20.67(0.07) | 21.15(0.09) |
| $\beta$-APF = 0.005 | 15.71(0.04) | 16.49(0.05) | 16.57(0.05) | 17.19(0.04) | 17.80(0.05) | 18.15(0.04) | 18.96(0.07) | 19.19(0.06) |
| $\beta$-APF = 0.01 | 15.69(0.04) | 16.31(0.04) | 16.31(0.04) | 16.85(0.04) | 17.47(0.05) | 17.66(0.05) | 18.46(0.05) | 18.66(0.05) |
| $\beta$-APF = 0.05 | 15.69(0.04) | 16.26(0.04) | 16.01(0.04) | 16.53(0.03) | 17.17(0.05) | 17.14(0.06) | 17.83(0.05) | 17.92(0.05) |
| $\beta$-APF = 0.1 | 15.84(0.04) | 16.46(0.05) | 16.16(0.04) | 16.56(0.04) | 17.30(0.05) | 17.16(0.04) | 17.89(0.05) | 18.09(0.05) |
| $\beta$-APF = 0.2 | 16.90(0.06) | 17.35(0.06) | 17.32(0.09) | 17.68(0.06) | 18.78(0.08) | 18.40(0.06) | 18.87(0.06) | 19.28(0.08) |

**Table 3:** Predictive results on the TAN example. The one step ahead predictive performance is measure by the median absolute error. The figures are averaged across 50 runs and the standard error on the average score is provided.

## G.3 London air quality experiment

**Table 4:** GP regression NMSE (higher is better) and 90% empirical coverage for the credible intervals of the posterior predictive distribution, on 100 runs. The **bold font** indicate the statistically significant best result according to the Wilcoxon signed-rank test. All presented results are statistically different from each other according to the test.

| Filter (Smoother) | median (IQR) | |
|---|---|---|
| | NMSE | EC |
| Kalman (RTS) | 0.144(0) | 0.685(0) |
| BPF (FFBS) | 0.116(0.015) | 0.650(0.020) |
| $(\beta = 0.005)$-BPF (FFBS) | 0.102(0.014) | 0.67(0.025) |
| $(\beta = 0.01)$-BPF (FFBS) | 0.077(0.007) | 0.705(0.015) |
| $(\beta = 0.05)$-BPF (FFBS) | 0.063(0.003) | 0.735(0.015) |
| $(\beta = 0.1)$-BPF (FFBS) | 0.061(0.003) | 0.760(0.015) |
| $(\beta = 0.2)$-BPF (FFBS) | **0.059(0.002)** | **0.803(0.020)** |

**Figure 26:** The GP fit on the measurement time series for one of the London air quality sensors. The top panel shows the posterior from the Kalman (RTS) smoothing. The middle panel shows the posterior from the BPF (FFBS). The bottom panel shows the posterior from the $\beta$-BPF (FFBS) for $\beta = 0.1$.

## Footnotes

[6]Bayesian additivity, also referred to as coherence says that applying a sequence of updates with subsets of the data should give rise to the same posterior distribution as single update employing all of the data.