[Reviews · NeurIPS 2020]

Review 1

Summary and Contributions: This paper considers an extension of the traditional general hidden Markov model (HMM) to the setting of misspecified likelihoods. The authors utilise the "general Bayesian inference" (GBI) framework to essentially extend the standard HMM to the setting where there is a mismatch between the model's likelihood function and the data generating mechanism. The typical approach for inferring the latent process from an HMM is to use a particle filter (i.e sequential Monte Carlo), which is also the approach taken here, albeit within the GBI framework, which results in two proposed algorithms, based on the beta divergence and which extend two traditional particle filter algorithms (bootstrap (BPF) and auxiliary (APF) particle filters). The authors provide supporting theory to show that their \beta-BPF convergences to the GBI posterior and a CLT result also holds. The paper has a simulation study which covers three models to assess the validity of their approach compared to standard bootstrap/auxiliary particle filtering algorithms.

Strengths: - This is a new contribution to HMMs and the SMC literature and I am not aware of similar work in this area being published which makes it novel. - A nice feature of this work is that when the likelihood is misspecified, e.g. the data are Student-t distributed rather than Gaussian, the user does not need to know the correct data generating distribution as the annealing aspect of \beta creates a simple way to use a robust likelihood function. - The paper offers a simple way of choosing the \beta parameter using the predictive distribution. - The paper offers some supporting theory and clear empirical evidence of the validity of their methodology. - I believe that the paper would be of interest to the NeurIPS community.

Weaknesses: - The authors choose to select \beta based on predictive accuracy. This is sensible, but what other approaches could also be used? What about model fit? And does it make sense to consider predictive accuracy on a separate training dataset? In the SMC community, people usually care more about the efficiency with which you can calculate the likelihood function (in order to estimate the parameters with particle MCMC), the accuracy of the filtered distribution, or ESS. Predictive accuracy is usually not a primary accuracy criterion so does it make sense to select \beta with this metric? From the simulation study, it appears that using predictive accuracy works well, but also seems to be consistently sub-optimal. Do you know why this is the case? - It's nice to see some supporting theoretical results for this work, but it seems to me that the results are just the standard convergence and CLT results that already exist in the particle filtering literature, but replacing \pi() with \pi^\beta(). It's not too surprising that these results hold for fixed \beta as your interest is now in estabashling convergence to \pi^beta rather than \pi, but is there any novelty in these results? The authors have clarified in their rebuttal that their results are given to demonstrate that existing work fits into their setting. - As discussed in Section 2.2, we are usually interested in \pi(\phi), but this GBI approach gives us \pi^\beta(\phi). Why are we interested in this quantity given that we don't really know what \beta equals? - The notation for \beta is a little counterintuitive, it states (in Section 2.1) that lim_{\beta->0}G^\beta()=g(), but in Section 3.2 it states "small values of \beta temper the likelihood," so you'd naturally think that \beta->1 would return to the original g_t(). Additionally, if the posterior is \pi^\beta(), then it would seem more natural (in terms of notation) that beta=1 would give you the original \pi(). You could probably change the notation so that beta->1 returns to the original posterior, but this might go against the convention in the GBI literature. In any case, I would suggest that the authors think a little bit about whether they can make the notation more intuitive for the reader. - The results all show how the \beta-BPF works in the setting of contaminated data, it would have been interesting to see the method applied in the non-contaminated setting. In other words, running Section 5.1 under the standard linear Gaussian setting and seeing if this method correctly infers that \beta is close to zero and still displays a high level of accuracy. I think this is quite important because the authors are suggesting that this is a "general" method that does not require the user to correctly specify the likelihood function, but what if the user does correctly specify the likelihood function? - The paper claims that the GBI framework (applied within the HMM setting) addresses the issue of model specification. However, it appears that what is being addressed is actually a restricted version of model specification. This approach handles situations where there are outliers in the data, but what if the data generating mechanism is a Gamma, or skewed-Normal distribution but the model likelihood is Gaussian? It appears that the \beta parameter acts to anneal the posterior, but is this not equivalent to increasing the variance in the Gaussian observation models considered in the experiments section? - It's interesting that parameter estimation is not considered. Given that there's a mismatch between the data generating mechanism and the model's likelihood function, it would mean that the parameters would be incorrectly estimated. However, would the estimated parameters not to some extent compensate for the misspecification (i.e. inflated variance parameter) and would this perhaps not lead to improved NMSE and EC?

Correctness: - To the best of my understanding, the paper is correct in its methodology and theory. The empirical results also appear correct, however, I have not run the code to confirm this, but I believe that the results are what you would expect to see from this work. - I couldn't find a proof of Corollary 1 in the Supplementary Material. If this result is trivial then it should be stated in the paper, or the proof referenced in other work.

Clarity: The paper is clear and very well-written. Sections 1 and 2 gave a good overview of existing work in this area and provided the necessary background information and notation needed to explain the authors' original contribution in Section 3.

Relation to Prior Work: The authors clearly outline in Section 1 how this work compares to similar work already available in the literature. The authors also empirically compare their method against competing approaches.

Reproducibility: Yes

Additional Feedback: - In Section 3.2 there is a short discussion about the integral in (9). It is stated that the integral can be solved analytically if g_t() is in the exponential family. For the experiments considered in Section 5 it appears that this is the case. However, the more interesting case is when this cannot be calculated analytically and it is noted in the paper that this can be estimated unbiasedly using the Poisson estimator. It would be useful if the authors could give some examples of popular HMM models where an approximation for (9) is required, e.g. what about stochastic volatility models? - In Section 5.3 it is stated that "..with known hyperparameters..." How are these known if the data is real and not simulated from a model? Small typos: Line 119 on page 3 - xbar_{t-1} should be xbar_t. Line 173 on page 5 - Typo "...can be seen as a standard SMC methods...." Line 243 on page 6 - Typo "...100x smaller that the Kalman...."


Review 2

Summary and Contributions: This work provides a generalised filtering framework based on Generalized Bayesian Inference (GBI) and Sequential Monte Carlo (SMC), specifically targeted to observation likelihood misspecification in general state-space Hidden Markov Models (HMMs). The proposed approach leverages: - GBI to accommodate more robust losses (to handle likelihood misspecification) - SMC methods to compute (generalized) filtering approximate densities The paper's contributions are: - A novel and useful Generalized SMC framework for robust filtering in the presence of likelihood misspecification - Theoretical guarantees on the convergence of the SMC-based posterior to the generalized posterior - Solid empirical evidence (synthetic scenarios and real-world data) of the benefits of the proposed approach

Strengths: The main strengths of the presented work are: - The proposal of simple, efficient and generalizable algorithms ($\beta$-BPF and $\beta$-APF) for robust filtering in the presence of likelihood misspecification - Providing posterior SMC converge guarantees for generalized posteriors based on the $\beta$ divergence. - Solid evidence of performance gains of $\beta$-PFs when compared to standard robust algorithms (on a variety of contamination settings).

Weaknesses: Some weaknesses/limitations of this work are: - The proposed method is not addressing the more general "problem of inference under full model misspecification": it only accounts for likelihood misspecification. In their rebuttal, the authors agree to change their terminology, as they "indeed focus on likelihood misspecification although the GBI framework of Bissiri, et. al. can in principle be used more widely, which we have now also highlighted." - The evaluation section could be slightly strengthen with some additional comparisons (see specific suggestions below). In their rebuttal, the authors have provided further evidence of their algorithm's performance, which I suggest they include in the updated manuscript (or clearly refer to, if left in appendix).

Correctness: The claims of this work are both based on theoretical and empirical evidence: - The posterior convergence guarantees are provided in Section 4, where the generalized filter is analyzed by accommodating the $\beta$-divergence based generalized likelihoods into classical SMC analysis - The empirical evaluation provided in Section 5 (based on 2 simulated yet challenging scenarios, and a real-world GP-based regression task) is sound: With the first two, the authors clearly convey the benefits of their proposed $\beta$ divergence based particle filtering: it provides robust inference in the presence of additive outliers. The real-data based GP regression showcases that the proposed $\beta$-PF outperforms other non-robust methods in the presence of outliers. In their rebuttal, the authors have addressed several of the proposed experiments, which I encourage them to include in the updated manuscript (or appendix): - What is the performance of a $\beta$-PF in the absence of outliers? As per results in their rebuttal, the performance depends on the used $\beta$ parameter, but their predictive selection finds moderate $\beta$ with near optimal performance. - How do the $\beta$-PFs compare to an "oracle"-PF that has knowledge of the true outlier model? The authors provide in their rebuttal evidence of how their predictive selection algorithms finds $\beta$ values leading to near optimal performance. - How do the $\beta$-PFs perform under other misspecified likelihood models: e.g., multiplicative noise, or other misspecified non-linear likelihood functions. The authors have clarified in their rebuttal that "more complex likelihood and contamination in the form of an asymmetric, multiplicative noise model was presented in Appendix D due to space constraints" (which I had previously missed). I encourage the authors to emphasize these results, either by promoting them to the updated manuscript or more clearly stating the relevance of this additional (non-additive) experiment

Clarity: The paper is very well written, it is very easy to read and follow. The authors provide thorough descriptions of the state-of-the art, their proposed method and the theoretical and empirical evidence to support their claims. These are some minor comments the authors should revise and correct, e.g.,: Line 22: "is the filtering distributions" -> "the filtering distributions" Line 155: "misspecitfied models" -> "misspecified models" Line 207: Add reference for "the Wilcoxon signed-rank test".

Relation to Prior Work: The authors have clearly presented the state of the art of the two key elements of their proposal (GBI and SMC), and have clearly stated their contribution: "a generalised filtering framework based on GBI, which tackles likelihood misspecification in general state-space HMMs."

Reproducibility: Yes

Additional Feedback:


Review 3

Summary and Contributions: The paper proposes a slight variation of the particle filter based ideas from Bissiri et. al. The idea is to account for uncertainties in the way the measurements are modelled by using a "more forgiving” distribution. This alters the algorithms slightly. The application in mind is a usual the hidden Markov model, which is indeed an important and highly useful model class. Straightforward adaptations of existing convergence results are provided.

Strengths: The paper in concerned with an important model class. The approach is theoretically well grounded.

Weaknesses: The main weakness of this work is due to its very limited novelty. The idea basically boils down to making use of a distribution with broader support in order to deal with uncertainty in the measurements, which is a rather standard approach for this case.

Correctness: The paper appears to be correct.

Clarity: The paper is well written and very easy to follow.

Relation to Prior Work: Yes. The amount of literature on this topic is massive, but I think that the authors have made a decent job in selecting relevant references.

Reproducibility: Yes

Additional Feedback:


Review 4

Summary and Contributions: The authors state that propose a new particle method for making inference in state space models "under likelihood misspecification".

Strengths: The paper is well-written and quite clear. The topic is relevant for the relevance to the NeurIPS community.

Weaknesses: In my opinion, the degree of novelty is very low. Related works (in my opinion) with the same goal or very related, are completely ignored.

Correctness: The material seems technically sound.

Clarity: The paper is well-written. This is main strength point.

Relation to Prior Work: In my opinion, the expression "....under likelihood misspecification" is very related to the "model selection" problem, and the "estimation of static parameters" in a dynamical model. In this sense, the state-of-the-art study is very poor; for instance, - N. Chopin, P. E. Jacob, O. Papaspiliopoulos. SMC2: an efficient algorithm for sequential analysis of state-space models. arXiv:1101.1528, 2013. - Dan Crisan, Joaquin Miguez. Nested particle filters for online parameter estimation in discrete-time state-space Markov models, arXiv:1308.1883, 2013. - L. Martino, J. Read, V. Elvira, F. Louzada, Cooperative Parallel Particle Filters for on-Line Model Selection and Applications to Urban Mobility, Digital Signal Processing Vol. 60, pp. 172-185, 2017. - I. Urteaga, M. F. Bugallo, and P. M. Djuric. Sequential Monte Carlo methods under model uncertainty, IEEE Statistical Signal Processing Workshop (SSP), pages 15, 2016. - C. M. Carvalho, M. S. Johannes, H. F. Lopes, and N. G. Polson. Particle Learning and Smoothing. Statist. Sci., Volume 25, Number 1 (2010), 88-106.

Reproducibility: Yes

Additional Feedback: The response of the authors helps me to understand the paper. This is a good work, in my opinion. ------------ The paper is well-written but, as I said above, my main concerns is the degree of novelty and the fact the authors ignore related works. Your problem can be easily solved considering parallel particle filtering each one addressing a different models or, similarly, as suggested in I. Urteaga, M. F. Bugallo, and P. M. Djuric. Sequential Monte Carlo methods under model uncertainty, IEEE Statistical Signal Processing Workshop (SSP), pages 15, 2016. See the other related references above.

[Author Response · NeurIPS 2020]



Wiener velocity: aggregate metrics for $p_c = 0.0$ — Influence on particle weight for different likelihoods — Wiener velocity: aggregate metrics for $p_c = 0.1$

We thank the reviewers for their time and constructive feedback. Our work provides a "new contribution to HMMs
and the SMC literature (R1)" in the form of "a novel and useful Generalised SMC framework for robust filtering in
the presence of likelihood misspecification (R2)" that is "theoretically well grounded (R3)" and of "relevance to the
NeurIPS community (R4)". Robust filtering in the M-open setting was an open problem that we have formally tackled.

**The proposed approach for selecting $\beta$ (R1)** is heuristic, but practically effective as the predictive performance is
closely related to the marginal likelihood which can also be used. Alternatives such as model fit can certainly be used as
well. We believe using a training set with the same statistical properties as the intended data makes sense practically
(and allows for online inference). We observed that using the same data did not alter performance. These points have
now been clarified in the manuscript.

**Interest in $\pi^\beta(\phi)$ (R1):** We have now clarified that, in the Bissiri et al. (2016) framework $\beta$ is a parameter of a specified
loss function: a subjective (generalised) Bayesian choice characterising confidence in model correctness. $\pi^\beta$ is then
considered the "correct" generalisation of the posterior within the GBI framework.

**Theory & Corollary 1 (R1)**: We agree that the arguments used here are standard and have now emphasised this in the
manuscript. The intention is to demonstrate that existing arguments hold in our setting, not to claim novelty. We have
added the proof of Corollary 1.

**$\beta$-divergence notation (R1):** We have now changed "small" to "positive" in 3.2 to convey the intended meaning.

**Additional evaluation (R1 & R2):** We thank the referees for their helpful suggestions, which have now been added
and emphasised in the paper. Please see some examples of additional results. 1) Non-contaminated regime on Wiener
velocity example (accidental omission) is shown on the left, where for moderate $\beta$ the performance is near optimal. 2)
Comparison to an oracle is one the right. 3) More complex likelihood and contamination in the form of an asymmetric,
multiplicative noise model was presented in Appendix D due to space constraints.

**Broadening the support of the likelihood (R1 & R3):** changing the likelihood (e.g. by increasing the measurement
variance) to decrease the influence of outliers suffers from drawbacks such as underestimating the influence of in-lying
observations and difficulty in interpreting results. Using the beta-divergence is principled and decreases the influence of
outliers while maintaining the influence on inliers (centre figure). We have included the influence function explanation.

**The relationship to parameter selection and model selection (R1 & R4)** have now been clearly signposted based
on Kantas et al. (2015) and R4's suggestions, including elaborating on parameter estimation in our setting and the
complications that arise in misspecified scenarios (cf. Brynjarsdóttir and O'Hagan, 2010). We consider challenging
M-open settings: we do not assume access to a family of models which includes the true generative model. Consequently,
model selection schemes cannot generally be used to correct for misspecification. We provide a formally justified
procedure for SMC algorithms that is forgiving to likelihood misspecification and allows for inferring posterior belief
distributions with desirable properties. This differs qualitatively from the listed literature in that the true model is not
available among our candidate models. In Urteaga, et. al. information from many candidate models is fused according to
their predictive performance, which is a pragmatic solution with good empirical performance on a univariate stochastic
volatility model, when a good suite of candidates is available. We view this as complementary to our work.

**Novelty (R3):** In our view, the principal novelty lies in interpreting filtering within the framework of GBI (a previously
unexplored direction) and hence arriving at a formally justified approach to robust filtering in the M-open setting.

**Examples of non-exponential family likelihoods (R1)**, such as student-t likelihoods have now been added and
discussed. Most stochastic volatility models could be addressed in closed form using properties of Gaussians.

**GP Hyperparameters (R1)** were selected by cross-validation. We have changed "known" to "fixed".

**Likelihood misspecification not model misspecification (R1 & R2):** We have changed the presentation and made the
change in terminology advocated by R2; we indeed focus on likelihood misspecification although the GBI framework
of Bissiri, et. al. can in principle be used more widely, which we have now also highlighted.

**Reproducibility (R4):** We have submitted the code as well as detailed descriptions of the experiments with the
supplementary material. If anything has been omitted, please inform us in review and we would be happy to include it.

[Meta-Review · NeurIPS 2020]

Reviewers agree that this is a clear contribution to the HMM and SMC set of methods that allows for robustness in the face of likelihood misspecification. Additionally, the method is both theoretically and empirically justified. While the main reviewer concern is novelty, there is agreement that the work is correct, thorough, and effective. An additional concern is the lack of clear attribution of theoretical results (e.g. proofs in the supplement) --- it would be useful for these statements to point to either the relevant work in the literature.